# Neutrophil extracellular trap-associated RNA and LL37 enable self-amplifying inflammation in psoriasis

Franziska Herster [1], Zsofia Bittner[1], Nathan K. Archer[2], Sabine Dickhöfer[1], David Eisel[1,14], Tatjana Eigenbrod[3], Thomas Knorpp[4], Nicole Schneiderhan-Marra [4], Markus W. Löffler [1,5,6], Hubert Kalbacher [7], Tim Vierbuchen [8], Holger Heine [8], Lloyd S. Miller[2], Dominik Hartl[9,10], Lukas Freund[11], Knut Schäkel [11], Martin Heister[12], Kamran Ghoreschi[12,13] & Alexander N.R. Weber [1]*

Psoriasis is an inflammatory skin disease with strong neutrophil (PMN) infiltration and high levels of the antimicrobial peptide, LL37. LL37 in complex with DNA and RNA is thought to initiate disease exacerbation via plasmacytoid dendritic cells. However, the source of nucleic acids supposed to start this initial inflammatory event remains unknown. We show here that primary murine and human PMNs mount a fulminant and self-propagating neutrophil extracellular trap (NET) and cytokine response, but independently of the canonical NET component, DNA. Unexpectedly, RNA, which is abundant in NETs and psoriatic but not healthy skin, in complex with LL37 triggered TLR8/TLR13-mediated cytokine and NET release by PMNs in vitro and in vivo. Transfer of NETs to naive human PMNs prompts additional NET release, promoting further inflammation. Our study thus uncovers a self-propagating vicious cycle contributing to chronic inflammation in psoriasis, and NET-associated RNA (naRNA) as a physiologically relevant NET component.

[1] Department of Immunology, University of Tübingen, Auf der Morgenstelle 15, 72076 Tübingen, Germany. [2] Department of Dermatology, Johns Hopkins University School of Medicine, Baltimore, MD 21231, USA. [3] Department of Infectious Diseases, Medical Microbiology and Hygiene, University Hospital Heidelberg, Im Neuenheimer Feld 324, 69120 Heidelberg, Germany. [4] NMI Natural and Medical Sciences Institute at the University of Tübingen, Markwiesenstr. 55, 72770 Reutlingen, Germany. [5] Department of General, Visceral and Transplant Surgery, University Hospital Tübingen, Hoppe-Seyler-Str. 3, 72076 Tübingen, Germany. [6] Department of Clinical Pharmacology, University Hospital Tübingen, Auf der Morgenstelle 8, 72076 Tübingen, Germany. [7] Interfaculty Institute of Biochemistry, University of Tübingen, Hoppe-Seyler-Str. 4, 72076 Tübingen, Germany. [8] Research Group Innate Immunity, Research Center Borstel, Leibniz Lung Center, Airway Research Center North (ARCN), Deutsches Zentrum für Lungenforschung (DZL), Parkallee 22, 23845 Borstel, Germany. [9] University Children's Hospital and Interdisciplinary Center for Infectious Diseases, University of Tübingen, Hoppe-Seyler-Str. 1, 72076 Tübingen, Germany. [10] Novartis Institutes for BioMedical Research (NIBR), Basel, Switzerland. [11] Department of Dermatology, University Hospital Heidelberg, Im Neuenheimer Feld 440, 69120 Heidelberg, Germany. [12] Department of Dermatology, University Hospital Tübingen, Liebermeisterstr. 25, 72076 Tübingen, Germany. [13] Department of Dermatology, Charité – Universitätsmedizin Berlin, Charitéplatz 1, 10117 Berlin, Germany. [14]Present address: David Eisel, BioNTech, An der Goldgrube 12, 55131 Mainz, Germany. *email: alexander.weber@uni-tuebingen.de

Psoriasis is an autoimmune disease of the skin with high incidence in Western countries (1.5–3%), causing high socioeconomic and disease burden with limited but increasing treatment options[1,2]. The most common form of psoriasis, plaque psoriasis, is characterized by epidermal hyperplasia due to keratinocyte (KC) hyper-proliferation, increased endothelial proliferation and an infiltrate of leukocytes, such as dendritic cells, T cells, and prominently, polymorphonuclear neutrophils (PMNs)[1]. The high and early accumulation of PMNs in psoriatic plaques and micro-abscesses is well documented, as well as an increase of PMNs in the circulation of psoriasis patients, but a specific causative role for PMNs in disease initiation or exacerbation has so far not been defined[3,4]. Although they are far less abundant in both blood and psoriatic skin, conversely, plasmacytoid dendritic cells (pDCs) have in the past received much attention in psoriasis. This interest has been fueled by the observations that pDCs can be activated by complexes consisting of either DNA or RNA with a antimicrobial self-peptide that is highly abundant in psoriatic skin, namely LL37: LL37 was shown to form complexes with DNA or RNA that resisted nuclease degradation, were readily taken up by pDCs, and triggered high interferon (IFN) α release from pDCs[5,6]. Expression of TLR7 (a pattern recognition receptor for single-stranded RNA) and TLR9 (a receptor for DNA) by pDCs was shown to be vital for the process in vitro.

However, as this scenario of pDC activation strictly requires the *prior* presence of DNA-LL37 and/or RNA-LL37 complexes, the above-described pDC-related mechanism does not qualify as an early initiating event in psoriasis and a process providing the three pDC-stimulating ingredients—LL37, DNA, and/or RNA—must be upstream. Unfortunately, the nature of this process has not been discovered to date.

Conversely to pDCs, PMNs can release DNA via so-called neutrophil extracellular trap (NET) formation, an activation-induced process leading to the extrusion of nuclear DNA[7]. In addition, cellular proteins are important components of NETs, and this includes LL37[8] for which PMNs are also the main producers in the skin[9]. LL37 is an amphipathic, positively-charged 37 amino acid peptide generated from a precursor protein, the cathelicidin hCAP18[1,10,11], that is stored in the secondary granules of PMNs, from where it can be released upon activation[9].

PMNs thus combine the abilities to release (i) DNA and (ii) LL37 as components of immunostimulatory ligand complexes via NET release, although this has not been firmly linked to psoriasis. They may themselves also sense such ligands via TLRs as expression of TLR8, another RNA sensor[12], and TLR9[13,14], but not TLR3 or TLR7[13], has been demonstrated but not functionally evaluated. We therefore hypothesized that PMNs may be the source of at least DNA and LL37 as immunostimulatory components and that LL37-mediated DNA sensing via TLRs in PMNs might initiate and fuel inflammatory cytokine production and thus inflammation and immune cell infiltration in psoriatic skin.

We here present experimental evidence that human primary PMNs do not sense DNA-LL37 but readily respond to RNA-LL37 complexes, leading to the release of a broad array of cytokines and chemokines and, importantly, NETs, via TLR8 (human) and TLR13 (mouse). Unexpectedly, these NETs contain RNA as a so-far unappreciated component, and they can propagate de novo NET release in naive human PMNs. PMNs, LL37 and, surprisingly, RNA are also highly abundant in psoriatic but not healthy skin, indicating that PMNs and NET-derived RNA-LL37 complexes may function as integral components of a self-propagating inflammatory cycle.

## Results

### LL37 promotes RNA uptake and PMN activation via TLRs.
Previous results indicated that primary human PMNs can respond to RNA and DNA when stimulated for >12 hours, albeit at much lower levels than when stimulated with the nucleoside analog TLR7/8 agonist, R848[14,15]. We sought to re-evaluate these findings using highly purified primary PMNs (gating strategy and activation status see Supplementary Fig. 1a) assayed within a short time period (4 h) that excludes secondary release effects, e.g., by apoptosis (Supplementary Fig. 1b). The TLR7/8 agonist R848, like the TLR4 agonist, LPS, elicited robust IL-8 release but only LPS triggered CD62L shedding; phospho-thioate (PTO) synthetic CpG ODN, a typical TLR9 agonist, also strongly activated IL-8 release and CD62L shedding (Fig. 1a, Supplementary Table 1). However, unmodified, natural phosphodiester DNA ODN or human genomic DNA elicited neither IL-8 release nor CD62L shedding, irrespective of whether they were complexed with LL37 (Fig. 1b). In contrast, in pDCs, LL37 binding of natural DNA triggered potent TLR9 responses[5]. In the absence of LL37, single-stranded synthetic RNA40 (henceforward referred to as 'RNA') barely caused IL-8 release when applied at equimolar concentrations with R848 (Fig. 1c). This suggests that on their own neither RNA nor DNA are able to trigger primary PMN responses. This may be due to the endosomal localization of TLR8 and 9, which R848 and CpG can apparently access, whereas RNA and normal DNA cannot[16]. However, robust IL-8 release and moderate CD62L shedding were observed when RNA was complexed with LL37 (Fig. 1c and cf. Fig. 2d). To check whether PMNs could engage RNA-LL37 complexes, live PMNs were seeded and RNA-LL37 complexes added for 20 min before fixing the cells for electron microscopy analysis. As shown in Fig. 1d, PMNs were found in proximity to fiber-like structures corresponding to RNA-LL37 complexes[17]. Flow cytometry using AlexaFluor (AF) 647-labeled RNA showed that LL37 also promoted the uptake of complexed RNA: >20% of PMNs incubated with labeled RNA in the presence of LL37 were AF647-positive within 60 min, compared to 5% in the absence of LL37 (Fig. 1e). ImageStream bright-field cytometry confirmed that this was not due to external/surface binding of labeled RNA but rather internalization (Fig. 1f, quantified in g). Bright-field fluorescence microscopy with Atto488-labeled LL37 showed co-localization of AF647-RNA and Atto488-LL37 in intracellular compartments (Fig. 1h), where endosomal TLRs are expressed[13]. To pharmacologically explore whether an endosomal TLR might be involved, we investigated the effect of chloroquine, a well-known inhibitor of endosomal TLRs that was non-toxic for PMNs (Supplementary Fig. 1c). Indeed IL-8 release from primary PMNs in response to RNA-LL37 and CpG ODN (positive control) was significantly reduced by chloroquine addition (Fig. 1i and Supplementary Fig. 1d). As chloroquine does not affect cytoplasmic RNA sensors (e.g., RIG-I)[18], this implicates an endosomal TLR-dependence in RNA-sensing. We next sought to check whether the observed effect on cytokine release extended to normal RNA from mammalian cells which may be more physiologically relevant in the context of psoriasis. As shown in Fig. 1j, human mRNA significantly induced IL-8 release, but only in combination with LL37. Since microbiome analyses have recently revealed alterations in staphylo- and streptococcal skin colonization in psoriasis patients[19] and bacterial RNA is an emerging immunostimulatory pattern[20], we also tested whether LL37 promoted S. aureus, i.e., bacterial, RNA (bRNA) activation of PMNs. Figure 1k shows that bRNA significantly stimulated IL-8 release similarly to RNA40 when complexed with LL37. Taken together this suggests that uptake of RNA is promoted by LL37 and directs RNA to intracellular compartments from which TLR sensing and activation leading to cytokine release can occur—both in response to mammalian and bacterial/foreign RNA.

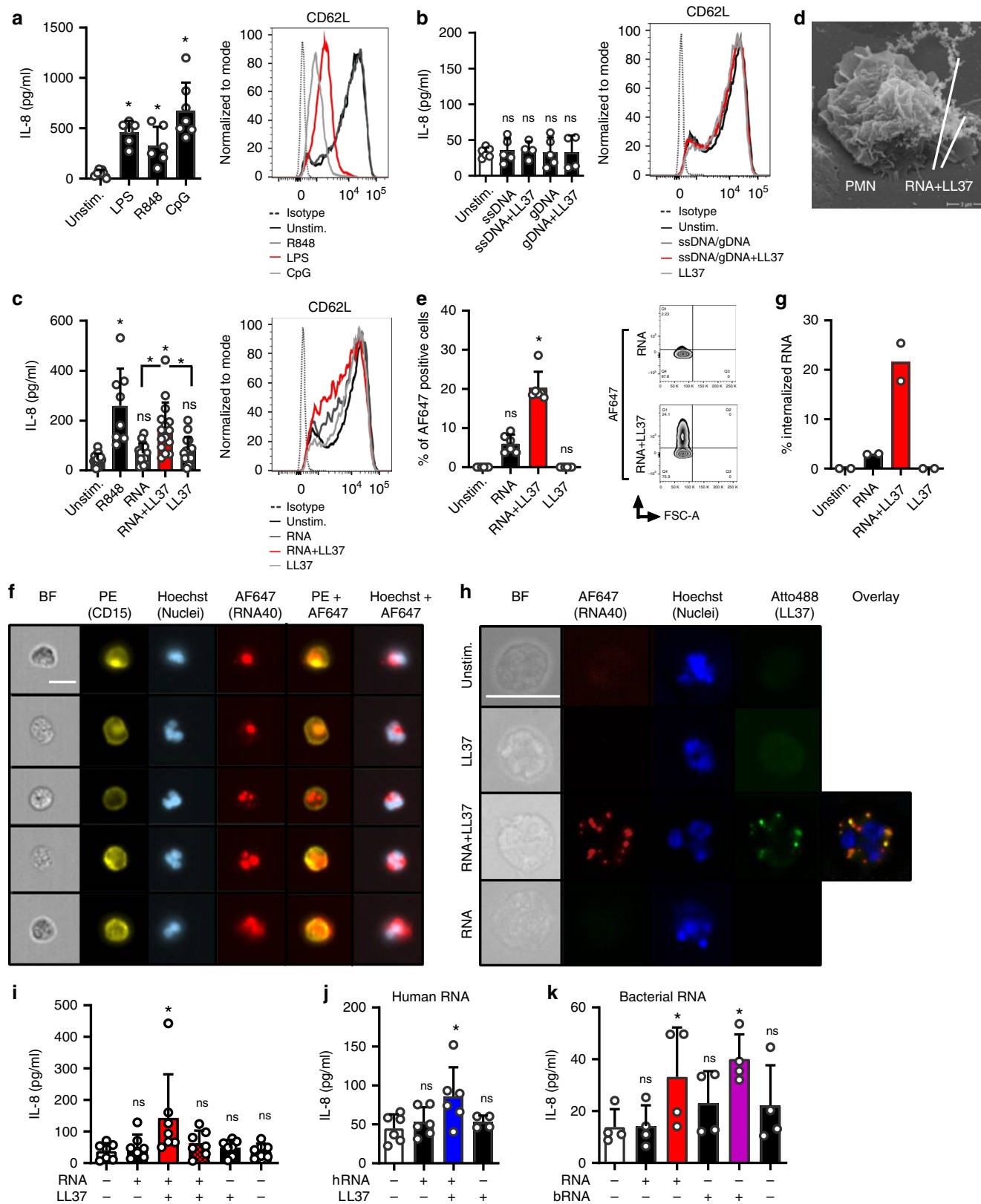

**RNA-LL37 complexes prompt cytokine and chemokine release.** To test whether RNA-LL37 also promoted the release of cytokines other than IL-8, we screened supernatants of stimulated PMNs by Luminex multiplex analysis and detected TNF, IL-1β, IL-6, and—unexpectedly—IL-16 (also known as Lymphocyte Chemoattractant Factor, LCF) and MIP-1β (also known as Chemokine (C-

C motif) Ligand 4, CCL4) primarily or exclusively in RNA-LL37-stimulated cell supernatants, respectively (Supplementary Fig. 2a, b). Despite considerable variation between donors, this was confirmed by cytometric bead array analyses (Fig. 2a–c) and ELISA (Fig. 2d, Supplementary Fig. 2c) in samples from more donors. The PMN response to RNA-LL37 complexes but not

**Fig. 1 Human PMNs take up and respond to RNA when complexed to LL37. a–c** IL-8 release (ELISA, 4 h stimulation) and CD62L shedding (flow cytometry, 1 or 2 h) by PMNs from healthy donors stimulated with LPS, R848 or CpG ODN (**a**, $n = 6$–7), ssDNA and genomic DNA (**b**, $n = 5$), or RNA40 (**c**, $n = 8$–15), with or without LL37. **d** Electron microscopy of PMNs incubated with RNA-LL37 for 20 min ($n = 2$). **e–h** FACS (**e**, $n = 6$), ImageStream cytometry images (**f**) and quantification (**g**, scale bar $= 10 \, \mu m$) or conventional bright-field microscopy (**h**, $n = 2$, scale bar $= 10 \, \mu m$) of PMNs incubated for 60 min with RNA-AF647 complexed with LL37 ($n = 6$). In (**f**) five selected cells from one representative donor are shown, in (**g**) the % of cells with 'internalized' features ($n = 2$, see Methods). In H unmodified LL37 was replaced with Atto488-labeled LL37, one representative donor shown. **i** as in (**c**) but including chloroquine (CQ) pre-incubation for 30 min ($n = 6$–7). **j, k** as in **c** but using total human RNA from HEK293T cells (**j**, $n = 4$–6) or bacterial RNA isolated from *S. aureus* (K, $n = 4$) with and without LL37. **a–c, e, f, i–k** represent combined data (mean + SD) from '$n$' biological replicates (each dot represents one donor). In **d, f**, and **h** representatives of '$n$' biological replicates (donors) is shown (mean + SD). $*p < 0.05$ according to one-way ANOVA with Dunnett's correction (**a, b, c, i, j**) or Friedman test with Dunn's correction (**e, k**). Source data is provided as a Source data file.

RNA was most prominent for MIP-1β (Fig. 2d). Since MIP-1β and IL-16 are known chemoattractants for CD4 T cells and a variety of other immune cells[21–23], we checked whether MIP-1β or IL-16, at the concentrations observed in this study, could influence the migration and infiltration of additional immune cells, which is a hallmark of psoriatic plaques[4]. Transwell experiments with peripheral blood mononuclear cells (PBMCs) from healthy human donors (upper well) and MIP-1β (30 and 150 pg/ml, Fig. 2e–g), IL-16 (300 and 1500 pg/ml, Supplementary Fig. 2d–f), or SDF-1α (100 ng/ml, Supplementary Fig. 2g–i) as a control (lower well) showed that the lowest concentration of both chemokines induced a donor-dependent increase in the number of CD3⁺CD4⁺ helper, CD3⁺CD8⁺ cytotoxic T cells as well as CD14⁺HLA-DR⁺ monocytes (Fig. 2e–g and Supplementary Fig. 2d–f). Possibly due to a non-linear dose–response relationship observed for some cytokines[24], higher concentrations had a weaker effect in some donors. Owing to the donor-to-donor variation generally observed throughout in the human system, only a non-significant, moderate effect was observed. We separately tested whether RNA or RNA-LL37 in the lower chamber had any direct influence on the migration of these cell populations and noted an unexpected and significant chemoattractive effect on CD4⁺ T cells, in response to RNA-LL37 complexes (Fig. 2h).

When repeating the above-described stimulation experiments with psoriasis PMNs compared to PMNs from healthy donors, MIP-1β release was significantly and two-fold higher in response to RNA-LL37 (Fig. 2i), but not to LPS (Supplementary Fig. 2j). Similar data were obtained for IL-8 (Supplementary Fig. 2k, l). The interesting observation that psoriasis PMNs constitutively showed an increased baseline release of LL37 compared to healthy donors (Fig. 2j) might explain, at least in part, why psoriasis PMNs respond more potently to RNA. We conclude that the combination of RNA with LL37 triggers the release of an extended array of pro-inflammatory cytokines and chemoattractants not triggered by RNA alone and that psoriasis PMNs respond more strongly to this stimulus.

**RNA-LL37 complexes trigger naRNA- and LL37-containing NETs.** Our data so far indicate that RNA and LL37 might contribute to cytokine-mediated inflammation and immune infiltration in neutrophil-containing skin lesions in psoriasis patients. Such neutrophil responses would probably be temporary, unless RNA-LL37 triggered the release of additional RNA and LL37. PMN NETs are known to contain extracellular DNA that acts as an immune stimulant of pDCs when complexed with LL37[25]. But DNA-LL37 cannot activate PMNs (cf. Fig. 1b) and whether RNA is released via NETs has not been shown. NETs would thus only propagate PMN activation in case they also contained RNA. We therefore next tested (i) whether RNA-LL37 induces NET release and (ii) whether NETs contain RNA. Interestingly, electron microscopy (Fig. 3a) and an elastase-based assay (see Methods) confirmed significant neutrophil elastase release, a hallmark of

NET release, in response to RNA-LL37 and PMA (positive control), but also with LL37 alone (Fig. 3b). However, fluorescence microscopy analysis of fixed PMN samples established that only RNA-LL37 complexes, and not LL37 alone, prompted the formation of LL37-positive NET-like structures (Fig. 3c). To examine whether the NETs also contained RNA, an RNA-selective fluorescent dye, SYTO RNAselect, was used to stain NETs. This staining showed released endogenous RNA (green) in both PMA and RNA-LL37-mediated NETs (Fig. 3c). Although the dye's specificity for RNA has been confirmed already[26] we showed that RNA staining, but not DNA staining, was sensitive to RNase A and is thus RNA-specific (Supplementary Fig. 3a, b). The presence of cellular RNA in NET structures was further evaluated by the use of a specific antibody against pseudo-uridine (ΨU), a nucleotide absent from both DNA and also the synthetic RNA40 used for LL37 complex formation and stimulation. Confirming the specificity of the antibody, the ΨU signal was completely RNase sensitive (Fig. 3d). Furthermore, using AF647- or AF488-labeled synthetic RNA for in vitro stimulation both the extracellular SYTO RNAselect and anti-ΨU signals could be unequivocally attributed to de novo released 'cellular RNA' and distinguished from exogenously added 'stimulant (synthetic) RNA' (Supplementary Fig. 3c, d). The same applied to LL37, using Atto488-LL37 (Supplementary Fig. 3e). To further exclude artefacts that might arise from staining fixed samples, RNA-LL37-mediated NET release was also analyzed and confirmed using live-cell time-lapse analysis of PMNs (Fig. 3e and Supplementary Movies 1–3, quantified in Fig. 3f). Interestingly, in PMNs captured on the verge of NET release, RNA staining in a granula-like fashion could be clearly observed (Supplementary Fig. 3f). Since increased levels of NETs containing DNA have already been reported in the blood and skin of psoriasis patients[27], we next investigated whether this was also the case for RNA. We therefore stained skin sections with SYTO RNAselect, anti-LL37 and anti-NE antibodies. Indeed, psoriatic skin was highly positive for LL37; interestingly, the RNA signal was also generally stronger when compared to healthy skin, using identical staining and microscopy settings (Fig. 3g). RNA and LL37 were frequently co-localized in samples with high PMN infiltration (anti-NE staining) (Fig. 3h and Supplementary Movie 4 to show co-localization in 3D). SYTO RNAselect specificity was also confirmed in skin sections using anti-ΨU staining with both stains clearly overlapping (Supplementary Fig. 4a). We next investigated whether PMN-derived NETs play a role in the imiquimod (IMQ, a TLR7 agonist)-induced experimental model of psoriasis in which topical application results in severe psoriasiform skin inflammation[28]. Immunofluorescence microscopy showed a co-localization of PMNs (MPO), NET DNA (histone H3), and RNA (using both SYTO RNAselect and anti-ΨU) in IMQ-treated but not naive mice ears (Supplementary Fig. S4b). Furthermore, in IMQ-treated mice ears NET-like structures could be detected at higher magnification (Supplementary Fig. 4c). Furthermore, injection of RNA-LL37 complexes into the ears of S100A8-EGFP mice or WT

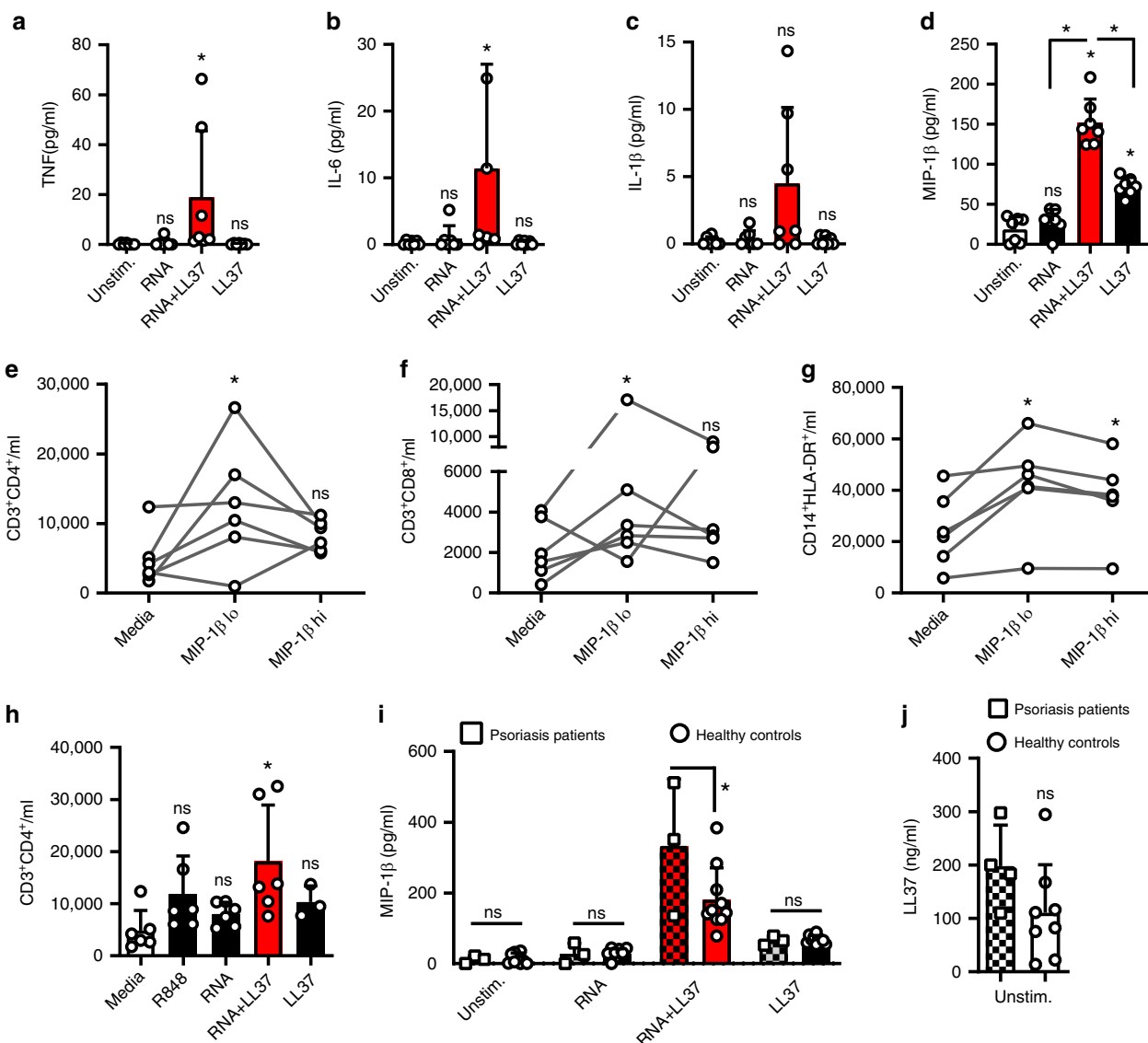

**Fig. 2 RNA-LL37 promote the release of cytokines and chemokines.** Cytometric bead array (**a**–**c**) or ELISA (**d**) for TNF (**a**), IL-6 (**b**), IL-1β (**c**), or MIP-1β (**d**) secreted from PMNs stimulated for 4 h as indicated ($n = 6$). Flow cytometric cell counts of migrated CD4+ T cells (**e**, **h**), CD8+ T cells (**f**) and CD14+ HLA-DR+ monocytes (**g**) quantified in transwell assays with total PBMCs in the upper and MIP-1β (30 and 150 pg/ml) (**e**-**g**, $n = 6$-7) or R848, RNA with and without LL37 (**h**, $n = 3$-7) in the lower chamber. **i** ELISA of MIP-1β secreted from psoriasis PMNs ($n = 3$ patients, 10 healthy controls, squares, chequered bars) or PMNs from sex and age-matched healthy donors. Only responses to treatment compared. **j** as in I but ELISA for LL37 ($n = 4$ patients, 7 healthy donors). Panels (**a**–**j**) represent combined data (mean + SD) from 'n' biological replicates (each dot represents one donor). *$p < 0.05$ according to Friedmann (**a**, **b**, **e**, **f**) or Kruskal–Wallis (**c**, **h**) test with Dunn's correction, or one-way ANOVA with Dunnett's (**d**, **g**, **h**) or Sidak (**i**) correction or Student's $t$-test (**j**). Source data is provided as a Source data file.

B6 mice (see methods) led to significant and immediate influx of PMNs within hours (Fig. 3i, k) and to significant ear swelling within 24 h (Fig. 3j). Furthermore, we observed extracellular RNA and citrullinated histone H3 (citH3, a NET marker) in close proximity to PMNs in the skin of mouse ears treated with RNA-LL37, but not PBS (Fig. 3k). In RNA-LL37 treated samples, a diffuse, overlapping and NET-like staining was observed for citH3 and ΨU in direct vicinity of MPO-positive PMN staining (Supplementary Fig. 4d), indicative of NET release in response to RNA-LL37 complexes in vivo. This suggests that RNA-LL37 complexes may act as physiologically relevant activators in psoriatic skin. Further, the ability of RNA-LL37 to drive primary PMNs to extrude NETs containing RNA in addition to DNA and LL37 may represent the so far unknown activating event upstream of pDC activation.

Based on these results it also appeared plausible that RNA-LL37 complexes might trigger a NET-mediated self-propagating inflammatory loop of repetitive PMN activation, in which NET material would act in a manner similar to RNA-LL37 and activate unstimulated/naive PMNs to undergo NET formation, leading to repetitive cycles of immune activation. Indeed, when we transferred NETs harvested from PMNs stimulated with PMA or RNA-LL37 complexes to unstimulated/naive PMNs, these responded with de novo NET release in turn (Fig. 3l and Supplementary Fig. 4e, quantified in Fig. 3m). Conversely, material harvested from LL37 or untreated PMNs, which did not contain NETs (therefore labeled 'mock NETs'), did not. Collectively, NETs generated by initial RNA-LL37 activation had the capacity to activate naive PMNs to extrude further DNA, RNA, and LL37, thus providing the requirements for a self-

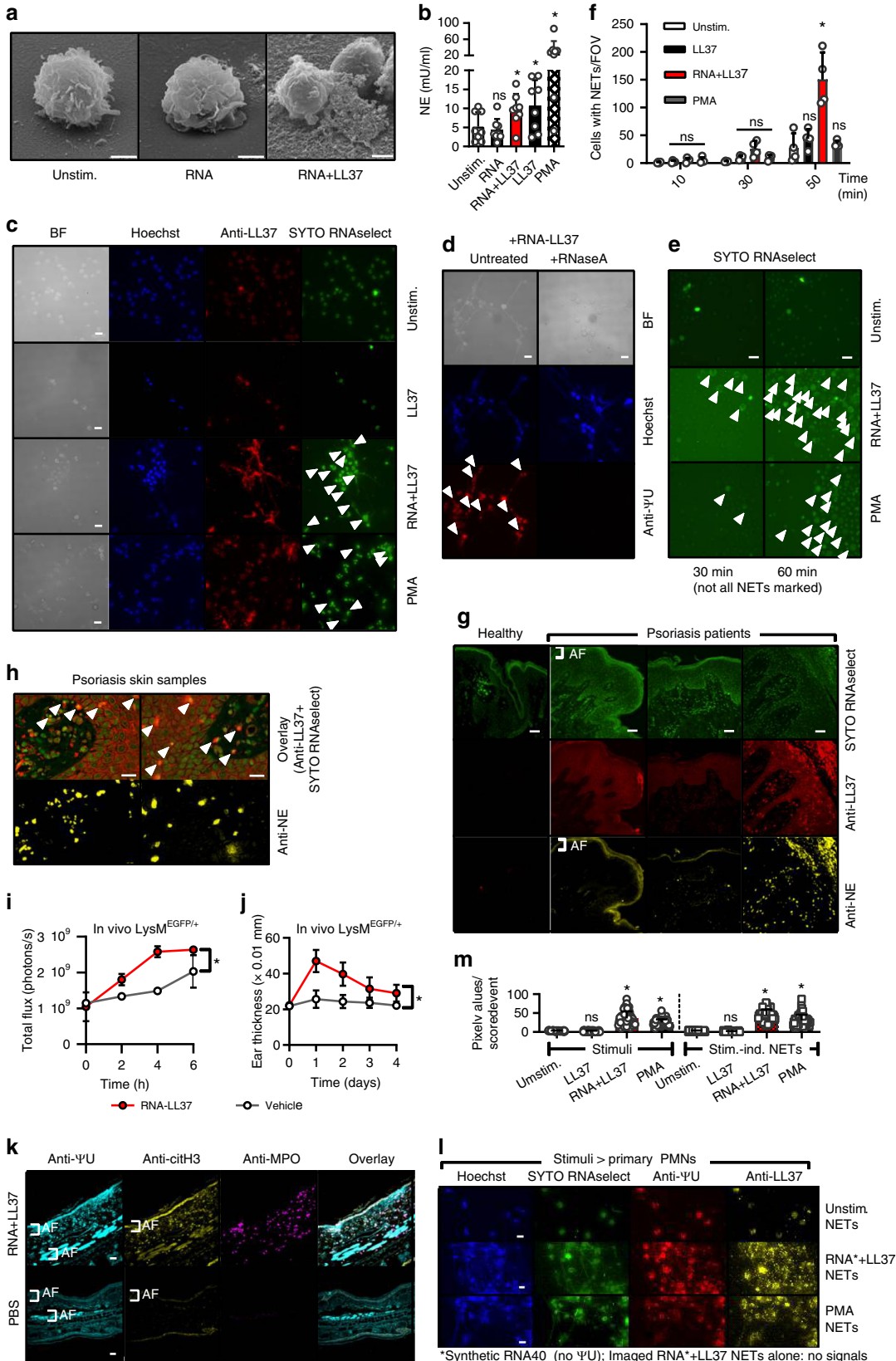

propagating feed forward loop of immune activation that might apply to skin PMNs. At the same time, we provide evidence for the so far unappreciated existence and physiological relevance of RNA in NETs, which we would like to term NET-associated RNA or naRNA (see Discussion).

**RNA-LL37 complexes and NETs activate PMNs via TLR8/TLR13.** In order to gain an insight which receptor mediates the observed effects of RNA-LL37 in primary PMNs, we first compared BM-PMNs from WT mice and mice deficient for *Unc93b1*, a critical chaperone of endosomal TLRs[29]. Evidently, not only the

**Fig. 3 RNA-LL37 trigger the release of NETs containing further RNA, DNA, and LL37. a** EM pictures from PMNs stimulated with RNA-LL37 ($n = 2$, scale bar = 2 μm). **b** Neutrophil elastase (NE) release from PMNs stimulated for 3 h ($n = 8$, each dot represents one donor). Fluorescence microscopy of fixed and Hoechst- anti-LL37- SYTO RNAselect- (**c**) and/or anti-ΨU-stained (**d**, with or without RNase A treatment) or live (**e**) PMNs stimulated as indicated ($n = 6$, scale bar = 10 μm or $n = 4$, scale bar = 20 μm). **f** Quantification of live-cell microscopy shown in Supplementary Movies 1–3. **g, h** Skin sections from healthy (**g**) or psoriasis- (**g, h**) affected skin ($n = 12$ patients and three healthy controls, scale bar = 20 μm). **i, j** RNA-LL37-induced PMN influx (**i**, $n = 3$ each, measured by GFP in vivo fluorescence) and ear swelling (**j**, $n = 6$ each, mm × 0.01) in LysM$^{EGFP/+}$ mice. **k** Ear sections from RNA-LL37 and PBS injected mice stained for ΨU, citH3 and MPO ($n = 6$, scale bar = 20 μm). **l** PMNs stimulated with transferred NETs ($n = 3$). **m** Quantification of (**l**). AF = autofluorescence. (**b**), (**f**), (**i**), (**j**), and (**m**) represent combined data (mean + SD) from '$n$' biological replicates. In (**a**), (**c–h**) and (**k, l**) representative samples of '$n$' replicates or donors are shown. Arrowheads indicate released RNA/NETs (**c–e**) or RNA-LL37 co-localization (**h**). *$p < 0.05$ according to one-way (**b**) or two-way ANOVA (**f, i, j**) with Dunnett's correction or Kruskal–Wallis test with Dunn's correction (**m**). Source data is provided as a Source data file.

cytokine response of BM-PMNs to bRNA-LL37 (and the control stimulus CpG) was strictly dependent on *Unc93b1* and thus endosomal TLRs (Fig. 4a); NET formation triggered by bRNA-LL37 complexes, but not the TLR-independent control stimulus PMA, was also *Unc93b1* dependent (Fig. 4b). Given that RNA is a known dual TLR7 and TLR8 agonist[30] and primary human PMNs do not show *TLR7* expression[13], we suspected TLR8 to mediate RNA-LL37-mediated PMN activation. Unfortunately, human primary PMNs are not amenable to RNAi or genetic editing. Therefore, we first tested the responsiveness of BM-PMNs from mice deficient for TLR13, the functional counterpart of human TLR8 and primary bRNA sensor in mice[20]. In murine PMNs both cytokine and NET responses to bRNA-LL37 were strictly dependent on TLR13 (Fig. 4a, c), whereas the negative controls LPS, heat-killed *E. coli* or PMA were not, respectively. Complementarily, the failure of *TLR8* CRISPR-deleted human BLaER1 monocytes[31] to respond to RNA-LL37 complexes (Fig. 4d) indicated a dependence of RNA-LL37 on TLR8 in the human system. Thus the stimulatory potential of (b) RNA-LL37 complexes can be considered to be dependent on the TLR8/ TLR13 RNA-sensing system[12].

**PMN cytokine and NET release by RNA-LL37 is blocked by iODNs.** Having identified TLR8 as the likely responsible receptor in humans, we next wondered whether PMN activation by RNA-LL37 could be blocked at the level of TLR activation by so-called inhibitory oligonucleotides (iODNs). IRS869, IRS661, and IRS954 represent such TLR7 and 9-inhibitory iODNs that were recently proposed for the treatment of another autoimmune disease, namely systemic lupus erythematosus (SLE). IRS546 was used as a non-inhibitory 'control' ODN[32–35] (see Methods). Given the similarities between endosomal TLRs[30], we speculated that iODNs with TLR7 and/or TLR9 antagonistic properties could potentially also block TLR8, which has not been investigated systematically. HEK293T cells do not express TLR7-9 but can be rendered responsive to their agonists by exogenously over-expressing TLR7, TLR8 or TLR9 upon transfection[30]. In TLR7, TLR8 or TLR9-transfected HEK293T cells, several iODNs blocked NF-κB activation in response to the control stimuli, R848 for TLR7 and 8 (Supplementary Fig. 5a, b) and CpG for TLR9 (Supplementary Fig. 5c). Importantly, for RNA complexed with DOTAP (a synthetic complexing agent promoting endosomal uptake similarly to LL37[36]), all three iODNs dose-dependently blocked TLR8-mediated NF-κB activation (Fig. 4e). The effects of all iODNs were specific for endosomal TLR signaling as NF-κB activities induced by TNF stimulation (Supplementary Fig. 5d) or MyD88 overexpression (Supplementary Fig. 5e) were unaffected. Having excluded a toxic effect on primary PMNs (Supplementary Fig. 5f), we investigated whether the observed effects could be transferred to endogenous TLR8 activation in primary PMNs, where LL37 as a physiologically relevant uptake reagent was again used. Here, iODNs IRS661 and IRS954, at nanomolar

concentrations, inhibited MIP-1β by primary PMNs in response to RNA-LL37 (Fig. 4f). Similar effects were observed for IL-8 (Supplementary Fig. 5g) and using chloroquine as a TLR8 inhibitor[37] to block MIP-1β release (Supplementary Fig. 5h). LPS-mediated IL-8 release was unaffected, confirming the specificity for RNA-LL37 complexes (Supplementary Fig. 5i). Importantly, NET formation by primary human PMNs could also be effectively blocked by nanomolar concentrations of IRS661 (Fig. 4g, quantified in Fig. 4h). These results are in agreement with the observed TLR8-dependence and iODN-mediated inhibition of NET formation in HIV infections[37] and indicate applicability to RNA-LL37 complexes. In conclusion, known pre-clinical iODNs are able to block RNA-LL37-mediated cytokine, chemokine and NET release.

## Discussion

Although systemic therapies with biologicals have brought benefit for many psoriasis patients in industrialized countries[2], psoriasis remains a significant health problem worldwide. One reason is that early triggers and mechanisms leading to chronic disease, exacerbation and flare-ups are not well understood. Previously, the effects of DNA-/RNA-LL37 complexes on pDCs were reported as an important mechanism in psoriasis inflammation[5,6]. However, since pDCs are only a minor cell population even in psoriatic skin and they cannot release nucleic acids or LL37, their contribution to cell-mediated inflammation and the provision of self-ligands had to be restricted to amplifying an already ongoing pathology, characterized by pDC-attracting chemokines and an existing presence of self-ligands in the skin. This begged the question of how nucleic acid-LL37 complexes may be released in the skin to trigger subsequent immune activation. Since PMNs can release nucleic acids, constitute the main sources of LL37 in humans and congregate in psoriatic skin, we chose to study PMN RNA-LL37-mediated responses after TLR8 was identified as an RNA receptor on PMNs[13]. We now provide evidence that PMNs are the likely early source of DNA, RNA, and LL37 that may be at the heart of inflammation in situ.

We here found that the combination of RNA and LL37 produces an inflammatory ligand that is not only sensed by PMNs but it also prompted the release of further components for additional RNA-LL37 and DNA-LL37 complexes from PMNs. Whereas for pDCs both "inflammatory ingredients", RNA or DNA and LL37, required the presence of another cellular source first, a small number of activated and NETing PMNs—by releasing both, LL37 and RNA/DNA—may spark potent, self-propagating inflammation amongst PMNs and in situ. Following an initial activation of PMNs by endogenous RNA from damaged skin cells (so-called Köbner phenomenon), immune activation may get out of control due to chemokine release and NET formation (and thus nucleic acid and LL37 extrusion[38]) which has been observed in the skin of psoriasis patients[27]. Although this will require further investigation, our data raise the intriguing

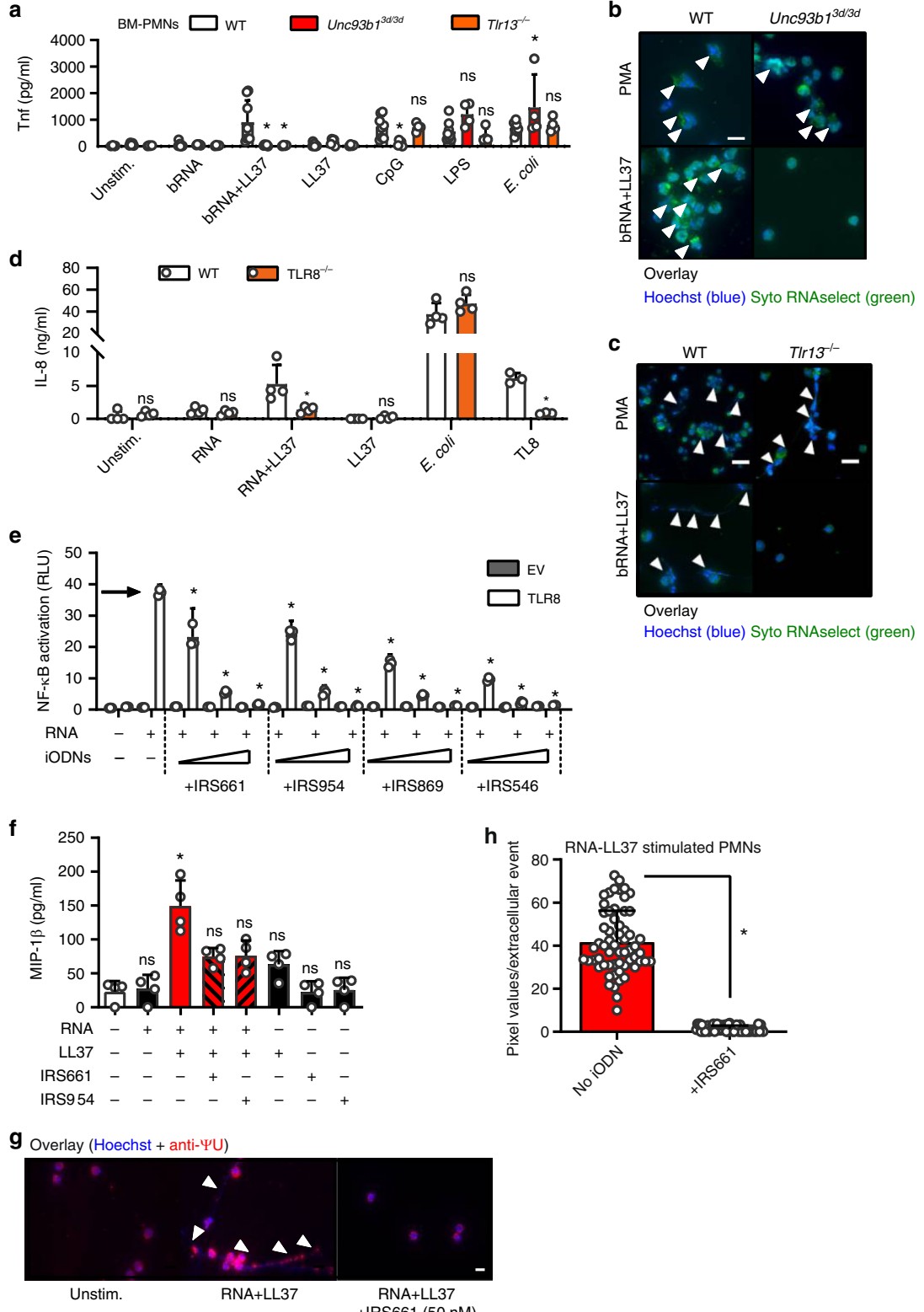

possibility that not only self-RNA, as previously proposed[6], but also foreign, e.g., bacterial RNA[20] (cf. Fig. 1k) or possibly fungal RNA may unfold immunostimulatory properties in the presence of LL37 triggered during minor skin injury. Although the concentration of the cytokines released after 4 h stimulation was moderate, release over longer time periods and the sheer number of PMNs found in psoriatic skin may nevertheless contribute substantially to local inflammation and attraction of additional leukocytes over time. The PMN-mediated ongoing release of NET DNA, RNA, and LL37 would eventually enable pDCs and other immune cells to join the vicious cycle of activation fueled by endogenous nucleic acids[5,6,39]. Although KC- and pDC-released type I IFNs contribute to psoriasis[40,41], TNF and IL-1β are additionally required for immune cell infiltration and T cell polarization, respectively[42], and RNA-LL37-mediated activation of PMNs may be the so far missing source of these cytokines.

**Fig. 4 RNA-LL37 effects on PMNs depend on TLR8 and are blocked by iODNs. a** Tnf release (ELISA, 5 h stimulation as indicated) by BM-PMNs from different mouse strains ($n = 4$–8 each). **b, c** Fluorescence microscopy of fixed and Hoechst and SYTO RNAselect-stained stimulated BM-PMNs from different mouse strains ($n = 4$ Tlr13$^{-/-}$, $n = 5$ Unc93b1$^{3d/3d}$, $n = 8$ WT; scale bar = 10 μm or $n = 4$, scale bar = 20 μm). **d** IL-8 and TNF release (ELISA, 18 h stimulation as indicated) by WT and *TLR8* CRISPR-edited BLaER1 cells ($n = 7$). **e** NF-κB dual luciferase reporter assay in HEK293T cells, transfected with NF-κB firefly luciferase reporter, *Renilla* control reporter and TLR8 plasmid and subsequently stimulated with RNA without (arrow) or with IRS661, IRS954, IRS869 and IRS546 (0.15–3.5 μM, $n = 2$ each). MIP-1β (**f**, $n = 4$) release from stimulated PMNs with or without IRS661 (1 nM) or IRS954 (50 nM) pre-incubation (30 min) quantified by ELISA. **g** Fluorescence microscopy of fixed and Hoechst and anti-ΨU-stained RNA-LL37-stimulated PMNs ($n = 3$, scale bar = 10 μm). **h** Quantification of (**g**). Panels (**a**), (**d**), (**e**), (**f**), and (**h**) represent combined data (mean + SD) from '$n$' biological replicates (each dot represents one mouse or donor). In (**b**), (**c**), (**e**), and (**g**) one representative of '$n$' replicates is shown (mean + SD of technical triplicates). *$p < 0.05$ according to two-way ANOVA with Dunnett correction for multiple testing (**e**, comparison against 'no inhibitor' condition, arrow), one-way ANOVA with Sidak correction (**a**, **d**) Friedmann test with Dunn correction (**f**) or Mann–Whitney test (**h**). Source data is provided as a Source data file.

Subsequent cytokine-mediated inflammation is known to favor hyper-proliferation[43], LL37 production[44], TLR9 responsiveness[45], and IFN production[46] in KCs, as well as T cell and monocyte attraction and polarization[42]. Our study thus raises the possibility that self-amplifying inflammation mediated by RNA and LL37 via TLRs recognition and NET formation of PMNs may represent an early and vital step in psoriasis development (Supplementary Fig. 6).

An unexpected finding that appears vital for this process and may have wide ramifications in many other NET-related processes, is the observation that RNA is an abundant component of mammalian NETs. More than 1000 publications on NETs have overlooked that NET-associated RNA (naRNA) is contained in NETs. Our data now demonstrate that naRNA is an abundant component of both RNA-LL37 and PMA-triggered NETs. The observation that it also triggers further immune activation of PMNs additionally demonstrates physiological relevance. NaRNA could thus represent as an additional immunostimulatory component within NETs, which has not been appreciated so far. Interestingly, eosinophils were reported to pre-store RNA within their granules[47], but with unknown significance. It can be assumed that RNA storage in granules is a general feature of granulocytes, in line with what we observed here (cf. Supplementary Fig. 3f). Of course, free RNA is not stable over long periods, but LL37 may confer protection not only for NETs in general[48] but also naRNA. NaRNA may thus be a common and functionally important NET component which remains to be explored further. Given that extracellular RNA can also amplify responses to other PRR ligands[49], the role of naRNA will thus be interesting to study in other NET-related diseases, e.g., SLE[50], atherosclerosis[51], and even cancer[52], where the focus so far has been exclusively on NET DNA. Furthermore, given that NET formation first and foremost has been described as an important host-defense mechanism[7], it remains to be investigated whether naRNA also executes or participates in antimicrobial activities.

In conclusion, our study offers insights into how PMN RNA-LL37 sensing and NET propagation may contribute to early disease development and, based on the observed effects of iODNs on PMN activation, warrants the further exploration of TLR pathways and PMNs as targets for restricting innate immune activation in psoriatic skin. In addition, our study also calls for an exploration of naRNA as an additional and intriguing component of NETs in psoriasis and other processes.

## Methods

**Reagents**. All chemicals were from Sigma unless otherwise stated. PRR agonists and LL37 were from Invivogen except RNA40 (iba Lifescience, normal and AF647- or AF488-labeled) and CpG PTO 2006 (TIB Molbiol), see Supplementary Tables 2 and 3. iODNs used in this study with their respective sequences are listed in Supplementary Table 3 and were from TIB Molbiol. As the 'control' ODN IRS546 blocked NF-κB activation in response to TLR7-9 stimulation, in our hands it did not represent a proper control unlike published[35]. Total human mRNA was isolated from HEK293T cells using the RNeasy kit on a QIAcube, both from Qiagen.

*S. aureus* RNA was isolated as described below. Genomic human DNA was isolated from whole blood using QIAamp DNA Blood Mini Kit from Qiagen (51106) and phosphodiester DNA from TIB Molbiol. LL37 was from InvivoGen (see Supplementary Table S2) and DOTAP from Roth, L787.2. LL37 was Atto488 (from Atto-Tec as a carboxy-reactive reagent)-labeled using standard procedures. For complex formation 5.8 μM RNA40 (~34.4 μg/ml and equimolar to R848 used in this setting), 1 μM ssDNA (~20 μg/ml and equimolar to CpG used in this setting, sequence see Supplementary Table 3), genomic DNA (20 μg/ml), or bacterial RNA (10 μg/ml) was mixed together with 10 μg LL37 (see Supplementary Table 2, Atto488 where indicated) and left for 1 h at RT. For experiments with BM-PMNs, 1.25 μg bacterial RNA was complexed with 2.5 μg LL37. For the RNA-only or LL37-only conditions, the same amounts and volumes were used replacing one of the constituents by sterile, endotoxin-free H$_2$O. Antibodies used for flow cytometry, ImageStream analysis or fluorescence microscopy are listed in Supplementary Table 4 as well as the recombinant cytokines used in this study. Constructs used for HEK293T transfection are listed in Supplementary Table 5.

**Preparation of bacterial RNA from *S. aureus***. For all experiments, RNA from *S. aureus* USA300 JE2 was used. On the first day, the bacterial culture was inoculated in suitable medium (using antibiotics if needed). The next day, the culture was diluted (1:100) and the OD was measured with 600 nm absorbance. The culture was further diluted to OD = 0.05, and incubated further with shaking at 37 °C until it reached an OD = 0.5. Five microliters of the samples were then harvested and spun down at $5000 \times g$ for 5 min at 4 °C. The supernatant was discarded and the pellet was resuspended in 1 ml Trizol (Invitrogen, 15596026) on ice and pipetted into freezing cups, containing zirconia silicon globules (Roth, N035.1). Then, the suspension was put on a shaker for 20 s at 6500 rpm before freezing at –80 °C. The samples were thawed at RT, then 200 μl chloroform was added and mixed for 30–60 s. After 3 min of incubation at RT, the samples were centrifuged at $12,000 \times g$ for 15 min at 4 °C. Thereafter, 500 μl isopropanol was pipetted into 1.5 ml RNase free Eppendorf tubes and the supernatant (~600 μl liquid phase) was added to the isopropanol. After mixing, and 10 min incubation at RT, the suspension was centrifuged at $12,000 \times g$ for 30 min at 4 °C. The supernatant was removed by pipetting, using filter tips and the pellet was washed with 500 μl of 70% ethanol and centrifuged again at $7500 \times g$ for 5 min at 4 °C. The supernatant was removed, leaving the pellet to dry. In the end, 50 μl of sodium citrate (1 mM, pH = 6.4 Ambion, AM7000) was added, incubated for 10 min at 55 °C on a heating block (the samples were thoroughly vortexed every 3–4 min) and frozen at −80 °C. The RNA concentration was determined with a Nanodrop Spectrophotometer before using for experiments.

**Study participants and sample acquisition**. All patients and healthy blood donors included in this study provided their written informed consent before study participation. Approval for use of their biomaterials was obtained by the local ethics committees at the University Hospitals of Tübingen (Chairman: Prof. Dr. med. D. Luft) and Heidelberg (Chairman: Prof. Dr. med. T. Strowitzki), in accordance with the principles laid down in the Declaration of Helsinki as well as applicable laws and regulations. All blood or skin samples from psoriasis patients (median age 41.8 years, PASI > 10, no systemic treatments at the time of blood/skin sampling) were obtained at the University Hospitals Tübingen or Heidelberg, Departments of Dermatology, respectively, and were processed simultaneously with samples from at least one healthy donor matched for age and sex (recruited at the University of Tübingen, Department of Immunology). Skin sections were from 12 patients with Plaque Psoriasis and 1 patient with Psoriasis guttate.

**Mice and isolation of bone-marrow derived PMNs (BM-PMNs)**. Unc93b1$^{3d/3d}$[53], *Tlr13*-deficient[54] mice (both C57BL/6 background) and WT C57BL/6 mice between 8 and 20 weeks of age were used in accordance with local institutional guidelines on animal experiments, regular hygiene monitoring, and specific locally approved protocols compliant with the German regulations of the Gesellschaft für Versuchstierkunde/Society for Laboratory Animal Science (GV-SOLAS) and the European Health Law of the Federation of Laboratory Animal Science Associations

(FELASA) for sacrificing and in vivo work. All mouse colonies were maintained in line with local regulatory guidelines and hygiene monitoring. *Unc93b1³ᵈ/³ᵈ* and WT C57BL/6 control mice were housed in the animal facilities at the Interfaculty Institute of Cell Biology, Tübingen, *Tlr13*-deficient and WT C57BL/6 control mice at the animal facilities at the Interfaculty Biomedical Facility, Heidelberg, respectively. Local state authorities for the approval of experimental protocols were the Regierungspräsidium Tübingen or Karlsruhe, respectively. Bone-marrow (BM)-PMNs were isolated from the bone marrow using magnetic separation (mouse Neutrophil isolation kit, Miltenyi Biotec, 130-097-658) and following the manufacturer's instructions. In total, $3 \times 10^6$ cells/ml PMNs were seeded and stimulation was carried out for 5 h at 37 °C and 5% $CO_2$. Thereafter supernatants were harvested and used for ELISA. For microscopy, the cells were stimulated for 16 h and subsequently stained.

**Neutrophil isolation and stimulation.** Whole blood (EDTA-anticoagulated) was diluted in PBS (Thermo Fisher, 14190-169), loaded on Ficoll (1.077 g/ml, Biocoll, ab211650) and centrifuged for 25 min at $509 \times g$ at 21 °C without brake. All layers were discarded after density gradient separation except for the erythrocyte-granulocyte pellet. Thereafter, erythrocyte lysis (using 1X ammonium chloride erythrocyte lysis buffer, see Supplementary Table 6) was performed twice (for 20 and 10 min) at 4 °C on a roller shaker. The remaining cell pellet was carefully resuspended in culture medium (RPMI culture medium (Sigma-Aldrich, R8758) + 10% FBS (heat inactivated, sterile filtered, TH Geyer, 11682258)) and $1.6 \times 10^6$ cells/ml were seeded (24-well plate, 96-well plate). After resting for 30 min at 37 °C, 5% $CO_2$, the cells were pre-treated with inhibitors (where indicated) for 30 min and subsequently stimulated with the indicated agonists for 4 h (for ELISA) or for 30 min to 3 h (for FACS analysis or microscopy).

**PBMC isolation.** Whole blood (EDTA anticoagulant) was diluted in PBS. After density gradient separation using Ficoll (described above), the PBMC layer was then carefully transferred into another reaction tube and diluted in PBS (1:1). The cell suspension was spun down at $645 \times g$ for 8 min. The cells were then washed twice more in PBS, resuspended in culture medium (RPMI + 10% FBS (heat inactivated) +1% L-glutamine), before counting and seeding.

**NET preparation.** Neutrophils from healthy donors were isolated as described above. They were seeded in 10 cm dishes at a cell density of $5 \times 10^6$ cells/ml at 37 °C and 5% $CO_2$. After resting the cells were stimulated with RNA-LL37 complex, LL37 alone, PMA (600 nM) or left unstimulated for 4 h. Supernatants were removed carefully and the adherent cells/NETs were washed three times with PBS. Then cells and NETs were scraped off the bottom of the dish and frozen in neutrophil culture medium at −80 °C.

**BLaER1 cells culture, transdifferentiation, and stimulation.** BLaER1 cells (WT and TLR8⁻/⁻, kindly provided by Tatjana Eigenbrod from Heidelberg) were cultured in culture medium (VLE-RPMI (Merck Biochrom, FG1415) + 10% FBS, 1% Pen/Strep (Gibco, 15140122), 1% sodium pyruvate (100 mM, Gibco, 11360070), 1% HEPES solution (Sigma, H0887)). After reaching a cell concentration not higher than $2 \times 10^6$ cells/ml, the cells were seeded in a 6-well plate ($1 \times 10^6$ cells/well) and transdifferentiated in culture medium adding 150 nM β-estradiol (Sigma-Aldrich, E2758), 10 ng/ml M-CSF (Peprotech, 300-25), and 10 ng/ml hIL-3 (Peprotech, 200-03) for 6 days including 2 medium changes (on day 2 and 5). On day 7 the adherent cells were counted again, seeded (96-well plate, $5 \times 10^4$ cells/well in culture medium) and left to rest for 1 h. Subsequently, they were stimulated for 18 h and the supernatants were harvested and collected for ELISA measurements. The transdifferentiation efficiency was verified by FACS analysis, using CD19, CD14, and CD11b as cell surface markers. CD19⁻CD14⁺CD11b⁺ were considered as monocyte-/macrophage-like cells.

**Flow cytometry.** After PMN isolation and stimulation, the purity and activation status of neutrophils was determined by flow cytometry. Two hundred microliters of the cell suspension was transferred into a 96-well plate (U-shape) and spun down for 5 min at $448 \times g$, 4 °C. FcR block was performed using pooled human serum diluted 1:10 in FACS buffer (PBS, 1 mM EDTA, 2% FBS heat inactivated) for 15 min at 4 °C. After washing, the samples were stained for ~20–30 min at 4 °C in the dark. Thereafter, fixation buffer (4% PFA in PBS) was added to the cell pellets for 10 min at RT in the dark. After an additional washing step, the cell pellets were resuspended in 150 μl FACS buffer. Measurements were performed on a FACS Canto II from BD Bioscience, Diva software. Analysis was performed using FlowJo V10 analysis software.

**ELISA.** Cytokines were determined in half-area plates (Greiner, Bio-one) using duplicates or triplicates and measuring with a standard plate reader. The assays were performed according to the manufacturer's instructions (Biolegend, R&D Systems), using appropriate dilutions of the supernatants. For LL37 determination a kit from HycultBiotech (HK321-02) was used following the manufacturer's instructions.

**ImageStream analysis.** ImageStream analysis was used to analyze internalization of RNA-LL37 complexes using spot-counts and tracking single cells. The cells were first seeded in a 96-well plate, $8 \times 10^6$ cells/ml, 125 μl per well. Subsequently, they were stimulated for 1 h with RNA-AF647 (IBA Technologies) and/or LL37-Atto488 (kindly provided by Hubert Kalbacher, University of Tübingen). FcR block and surface staining (here CD15 PE) was performed as described above. After fixation, the cells were permeabilized with 0.05% Saponin (Applichem, A4518.0100) for 15 min at RT in the dark. After washing, nuclei were stained with Hoechst 33342 (Sigma, B2261, 1 μg/ml) for 5 min at RT in the dark, washed and resuspended in 50 μl FACS buffer and transferred into a 1.5 ml Eppendorf tube. At least 10,000 cells were acquired for each sample with ×40 magnification using an ImageStream X MKII with the INSPIRE instrument controller software (Merck-Millipore/Amnis). Data were analyzed with IDEAS Image analysis software. All samples were gated on single cells in focus.

**Fluorescence microscopy of fixed neutrophils.** The cells were seeded in a 96-well plate at $1.6 \times 10^6$ cells/ml, 125 μl per well. Subsequently they were stimulated with the complexes for 30 min and 1 h using RNA-AF647/AF488 and/or LL37-Atto488. FcR block, staining, fixation and permeabilization were performed as for Flow cytometry. The cell pellets were resuspended in 50–100 μl FACS buffer. Forty microliters of the cell suspension was pipetted on a Poly-L-Lysine-coated coverslip (Corning, 734-1005) and the cells were left to attach for 1 h in the dark at RT. ProLong Diamond Antifade (Thermo Fisher, P36965) was used to mount the coverslips on uncoated microscopy slides. For NET analysis PMNs were seeded in 24-well plates, containing Poly-L-Lysine-coated coverslips and stimulated with RNA-LL37 complexes or PMA (600 nM) for 3 h. NETs were fixed and stained using the protocol from Brinkmann et al.[55]. Where indicated, 100 μg/ml RNase A (DNase, protease-free, Thermo Fisher, EN0531) was added after fixation and incubated overnight at 37 °C. RNA was stained using SYTO RNAselect Green fluorescent dye (Thermo Fisher, 50 μM) or anti-ΨU antibody (see Supplementary Table 4) and nuclear DNA was stained with Hoechst 33342 (Thermo Fisher, 1 μg/ml). LL37 and PMNs were visualized using an unconjugated rabbit anti-LL37 antibody or unconjugated mouse anti-Neutrophil Elastase (NE) with subsequent staining with an AF647-conjugated anti-rabbit or an AF594-conjugated anti-mouse antibody, respectively (see Supplementary Table 4), after blocking with pooled human serum (1:10 in PBS). Secondary antibodies alone did not yield any significant staining. The slides were left to dry overnight at RT in the dark and were then stored at 4 °C before microscopy. The measurements were conducted with a Nikon Ti2 eclipse (×100 magnification) and the analysis was performed using Fiji analysis software.

**Fluorescence microscopy of tissue samples.** Skin samples from psoriasis patients with a PASI ≥ 10 and without systemic treatment, and healthy skin samples paraffin-embedded according to standard procedures were deparaffinized and rehydrated using Roti Histol (Roth, 6640.1) and decreasing concentrations of ethanol (100, 95, 80, and 70%). After rinsing in dd$H_2O$, antigen retrieval was performed by boiling for 10–20 min in citrate buffer (0.1 M, pH = 6). The skin tissue was then washed three times for 5 min with PBS. Blocking was performed using pooled human serum (1:10 in PBS) for 30 min at RT. The primary antibody was added either overnight at 4 °C or for 1 h at RT. After three washes, the secondary antibody was added for 30 min at RT in the dark. After another three washes, SYTO RNAselect Green fluorescent dye (Thermo Fisher, 50 μM) was added for 40 min at RT in the dark. Thereafter, the samples were washed again and Hoechst 33342 (Thermo Fisher, 1 μg/ml) was added for 5 min. Then the last three washes were performed before using ProLong Diamond Antifade (Thermo Fisher, P36965) for mounting. The samples were left to dry overnight at RT in the dark before being used for microscopy or stored at 4 °C. The specimens were analyzed on a Nikon Ti2 eclipse bright-field fluorescence microscope (×10–60 magnification) and the analysis was performed using Fiji analysis software. Autofluorescence in multiple channels typical, e.g., for the stratum corneum was labeled "AF".

**Live-cell imaging of primary neutrophils.** Human neutrophils were isolated by magnetic separation using MACSxpress whole blood neutrophil isolation kit (Miltenyi Biotec, 130-104-434) and $1.6 \times 10^6$ cells/ml were seeded into a micro-insert 4-well dish (Ibidi, 80406). Hoechst 33342 (1 μg/ml) and SYTO RNAselect Green fluorescent dye (50 μM) were added to the cells and incubated for 20 min at 37 °C, 5% $CO_2$. Live-cell imaging was performed by using Nikon Ti2 eclipse bright-field microscope (×40 magnification) including a $CO_2$–$O_2$ controller from Okolab. Measurements were started immediately after adding of stimuli. Time-lapse analysis was performed by taking pictures every 3 min for at least 2 h. Image analysis was performed using NIS Elements from Nikon and Fiji analysis software.

**Luminex cytokine multiplex analysis.** All samples were stored at −70 °C until testing. The samples were thawed at room temperature, vortexed, spun at $18,000 \times g$ for one min to remove debris and the required sample volumes were removed for multiplex analysis according to the manufacturer's recommendations. The samples were successively incubated with the capture microspheres, a multiplexed cocktail of biotinylated, reporter antibodies, and a streptavidin-phycoerythrin (PE)

solution. Analysis was performed on a Luminex 100/200 instrument and the resulting data were interpreted using proprietary data analysis software (Myriad RBM). Analyte concentrations were determined using 4 and 5 parameter, weighted and non-weighted curve fitting algorithms included in the data analysis package.

**Cytometric bead array**. A cytometric bead array was performed using the "Human inflammatory cytokine kit" from BD Bioscience (551811) and following the manufacturer's instruction. Twenty-five microliters of samples and standards were added to 25 μl of the capturing bead mixture. In addition, 25 μl of PE detection reagent was added to all tubes and incubated for 3 h at RT in the dark. Thereafter, 1 ml of wash buffer was added to each tube and centrifuged at $200 \times g$ for 5 min. The supernatant was carefully removed and the pellet was resuspended in 300 μl wash buffer. Measurements were performed with the FACS Canto II from BD Bioscience operated using Diva software. Analysis was performed with Soft Flow FCAP Array v3 analysis software from BD Bioscience.

**Transwell experiments**. Transwell inserts (using polycarbonate, 24-well plates, 3 μm pores, Corning, 734-1570) were loaded with 100 μl of PBMC suspension ($0.8 \times 10^6$ cells/insert). In the lower chamber, media containing stimuli only or media containing only MIP-1β (30 and 150 pg/ml), IL-16 (300 and 1500 pg/ml) or SDF-1α (control, 100 ng/ml) were added. After 4 h, the lower compartment was harvested and FACS staining was performed as described above. The total number of migrated cells was acquired using counting beads (Biolegend, 424902) on a FACS Canto II (BD Bioscience) with Diva software. Analysis was performed using FlowJo V10 analysis software.

**Neutrophil elastase NET formation assay**. Neutrophil extracellular trap formation was determined using the colorimetric NET formation Assay Kit from Cayman Chemicals based on the enzymatic activity of NET-associated neutrophil elastase. PMNs from various healthy donors were isolated as described above and stimulated with RNA-LL37 complex, or PMA and a calcium ionophore (A-23187) as positive controls for 1 or 3 h. The assay was performed following the manufacturer's instructions. The absorbance was then measured at 405 nm using a standard plate reader.

**Transient transfection of HEK293T cells**. HEK293T cells were transiently transfected using the CaPO$_4$ method: Cells were seeded in 24-well plates at a density of $1.4 \times 10^6$ cells/ml, 2–3 h prior to transfection. For the transfection of one well, 310 ng of plasmid DNA (100 ng TLR plasmid, 100 ng firefly luciferase NF-κB reporter, 10 ng *Renilla* luciferase control reporter, and 100 ng EGFP plasmid) was mixed with 1.2 μl of a 2 M CaCl$_2$ solution and filled up with sterile endotoxin-free H$_2$O to obtain a total reaction volume of 10 μl. After the addition of 10 μl of 2X HBS solution (50 mM HEPES (pH 7.05), 10 mM KCl, 12 mM Glucose, 1.5 mM Na$_2$HPO$_4$), the mixture was then added to the cell suspension. As negative controls, TLR coding plasmids were replaced by empty vectors carrying the appropriate backbone of the TLR plasmids. After the addition of the transfection complexes, the cells were incubated either for 24 h followed by stimulation, or kept for 48 h without stimulation (MyD88 expression). For stimulation, the media was aspirated and replaced by fresh growth medium in which TNFα or the different TLR ligands (R848, CpG, RNA40) with or without IRS were diluted to appropriate concentrations. TLR8 activation with RNA40 was facilitated by complexation of the RNA with DOTAP (L787.1, Roth). RNA40 and DOTAP were first diluted in 1X HBS separately. Next, RNA40/HBS was diluted 1:3 in DOTAP/HBS. The solution was carefully mixed by pipetting up and down. After 15 min of incubation at RT, the mixture was 1:6.7 diluted in growth medium (with or without IRS) and finally dispensed (500 μl/well) into the wells containing transfected HEK293T cells. Each tested condition was measured in triplicates. The cells were stimulated and inhibited for 24 h at 37 °C.

**Dual luciferase reporter assay**. After checking transfection efficiency via EGFP fluorescence microscopy, HEK293T supernatants were aspirated and 60 μl of 1X passive lysis buffer (E194A, Promega) added per well. The plate was incubated for 15 min at RT on the plate shaker and subsequently stored at −80 °C for at least 15 min to facilitate complete cell lysis. After thawing, 60 μl were transferred into a V-bottom 96-well plate and centrifuged for 10 min at 2500 rpm and 4 °C to pellet cell debris. Ten microliters of supernatant were then transferred into a white microplate and each condition was measured in triplicates using the FLUOstar OPTIMA device (BMG Labtech). Firefly and Renilla luciferase activity were determined using the Promega Dual luciferase kit. Both enzyme activities were measured for 12.5 s with 24 intervals of 0.5 s, respectively. The data were analyzed by calculating the ratio of the two measured signals, thereby normalizing each firefly luciferase signal to its corresponding Renilla luciferase signal. The ratios were represented as the relative light units (RLU) of NF-κB activation.

**Imiquimod model of psoriatic skin inflammation**. Prior and after daily topical application of 5% imiquimod cream to the ears of anesthetized (2% isoflurane) C57BL/6 mice full thickness ear skin was excised on day 5, fixed in 10% formalin

and paraffin-embedded. Skin cross-sections (4 μm) were stained according to standard procedures as indicated.

**Neutrophil infiltration: in vivo fluorescence imaging**. Female LysM$^{EGFP/+}$ mice (C57BL/6 genetic background[56], bred and maintained under specific pathogen-free conditions with air-isolated cages at an American Association for the Accreditation of Laboratory Animal Care–accredited animal facility at Johns Hopkins University) were housed in the animal facility at the Johns Hopkins University School of Medicine. These experiments were conducted according to local official policies, procedures, and animal experiment approvals granted by Johns Hopkins University IASCUC (approval M018M329). Sex- and age-matched 6- to 8-week-old mice were used for each experiment. For RNA-LL37 administration, mice ears were injected intradermally with 20 μL of RNA-LL37 complex or vehicle (endotoxin-free water) on the same day as complex preparation. LysEGFP mice were then anesthetized with inhalation isoflurane and in vivo fluorescence imaging was performed using the IVIS Lumina II imaging system (Caliper). EGFP fluorescence was measured using: excitation (465 nm), emission (515–575 nm), and exposure time (0.5 s). Data are quantified as total radiant efficiency ([photons/s]/[μW/cm$^2$]) within a circular region of interest using Living Image software (Caliper). Ear thickness was additionally measured using a manual caliper.

**Neutrophil infiltration: ex vivo immunofluorescence analysis**. C57BL/6 WT mice were bred, maintained and handled according to local official policies (see above), procedures and animal experiment approvals compliant with the German regulations of the Gesellschaft für Versuchstierkunde/Society for Laboratory Animal Science (GV-SOLAS) and the European Health Law of the Federation of Laboratory Animal Science Associations (FELASA) were granted by the local state authority (Regierungspräsidium Tübingen, Germany, approval IM 01/19G). In all, 40 ng/μl bacterial RNA (isolated from *S. aureus USA300 JE2*) was complexed with 80 ng/μl LL37 for 1 h at RT. Two microliters of the complex and PBS (complex right ear, PBS left ear) were injected intradermally into the ears of WT C57BL/6 mice between 8 and 16 weeks of age[57]. 600 nM PMA served as a control stimulus. Seven hours after injection the mice were sacrificed using CO$_2$, followed by cervical dislocation. The ears were cut off, dehydrated, embedded into paraffin, cut (thickness = 3 μm) and mounted on SuperFrost plus slides (Langenbrinck, 30-0060) using standard protocols. Then the tissue samples were deparaffinized, rehydrated, antigen retrieval was performed and stained as indicated and mounted using ProLong Diamond Antifade mounting solution and left to dry at RT o/n in the dark. The specimens were analyzed using a Nikon Ti2 eclipse microscope (×40 magnification using immersion oil).

**Statistics**. Experimental data were analyzed using Excel 2010 (Microsoft) and/or GraphPad Prism 7 or 8, microscopy data with ImageJ/Fiji or Nikon's NIS Elements II, flow cytometry data with FlowJo V10. In case of extreme values, outliers were statistically identified using the ROUT method at high (0.5%) stringency. Normal distribution in each group was always tested using the Shapiro–Wilk test first for the subsequent choice of a parametric (ANOVA, Student's *t*-test) or non-parametric (e.g., Friedman, Mann–Whitney U or Wilcoxon) test. *p*-values ($\alpha = 0.05$) were then calculated and multiple testing was corrected for in Prism, as indicated in the figure legends. Values < 0.05 were generally considered statistically as significant and denoted by * throughout. Comparisons made to unstimulated control, unless indicated otherwise, denoted by brackets.

**Reporting summary**. Further information on research design is available in the Nature Research Reporting Summary linked to this article.

## Data availability
The authors declare that all data supporting the findings of this study are available within the paper, its supplementary information files and/or the Source Data file. All unique materials used are readily available from the authors upon reasonable request or from standard commercial sources as specified in the methods section.

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

## Acknowledgements

We thank S. Pöschel, J. Berger, S. Haen, K. Preissner, O. Sorensen, S. Kohler, N. Korn, and A. Dalpke for provision of reagents, samples, technical support and/or helpful discussions, and all healthy donors and patients for participation in our study. This study was supported by the Deutsche Forschungsgemeinschaft (German Research Foundation, DFG) CRC TR156 "The skin as an immune sensor and effector organ—Orchestrating local and systemic immunity", the University of Tübingen and the University Hospital Tübingen.

## Author contributions

F.H., Z.B., N.K.A., S.D., D.E., T.K. and N.S.M. performed experiments; F.H., Z.B., N.A., S.D., D.E., N.S.M. and A.N.R.W. analyzed data; M.H., H.K., T.V., H.H., M.W.L., L.F., K.S., L.M., K.G. and T.E. were involved in sample and reagent acquisition; F.H., D.H., K.G. and T.E. contributed to the conceptual development of the study; F.H. and A.N.R.W. wrote the paper and all authors commented on an revised paper; A.N.R.W. coordinated and supervised the entire study.

## Competing interests

The authors declare no competing interests.
