## [Peer Review File · Nature Communications]

Reviewers' comments:

Reviewer #1, expert on neutrophils (Remarks to the Author):

The manuscript by Herster et al. reports a series of experiments that the authors have performed in order to better understand the underlying molecular basis for psoriatic skin lesions (PSL). It has been known that PMN infiltrate PSL and that the sites may also show accumulations of LL-37, a peptide that is produced and released by PMN. The LL-37 peptide was known to bind DNA and RNA and therefore the authors asked whether the combination of LL-37 and RNA could stimulate an inflammatory response that would involve cytokine secretion and neutrophil activation.

The authors present Figure 1) to argue that RNA-LL37 stimulates IL-8 cytokine release and the shedding of 62L marker from healthy neutrophils. However, the response is small (less than an RNA oligo R848 or other positive controls can achieve) and the CD62L shedding is a relatively subtle effect. So, the initial data in this report are nearly at background levels except for the uptake of label by PMN, which was anticipated from the known binding of LL37 to RNA.

In Figure 2), the secretion of TNF, IL-1 and IL-6, MIP1b and IL-16 are shown but the samples indicate that only certain few donors have a strong response, whereas most samples are not above background.

In Figure 3), samples of PMN from psoriasis patients are assayed but again the results are not very striking.

The Figure in which RNA is measured by SYTO RNA-Select, concern that some of the signal is contributed by DNA cannot be avoided because the manufacturer indicates a weak binding to DNA.

In sum, the study is somewhat original but the overall purpose, to provide a major pathogenic mechanism for psoriasis, is not fully achieved. Instead, the study is consistent with previous knowledge and demonstrates a weak, but potentially meaningful, involvement of RNA-peptide complexes as further stimuli to amplify the innate immune response in psoriasis.

Reviewer #2, expert on psoriasis (Remarks to the Author):

"RNA-antimicrobial peptide complexes fuel a TLR- and NETosis-mediated inflammatory cycle of neutrophil activation in psoriasis". I have the following comments.

1. The chloroquine inhibition of response to RNA-LL37 – led the authors to conclude that TLR8 was the receptor mediating the observed effects in PMNs. The authors also show co-localization of RNA40 with TLR8 in PMNs (Figure 1H). The role of TLR8 in this process is of interest but I do think the authors need to demonstrate this more conclusively by using siRNA approaches etc. I realize that PMNs can be hard to work with but I think this is essential, particularly as some groups have shown that chloroquine may augment activation of synthetic compounds such as R848 (Kuznik A, JI 2011).

Therefore, the role of chloroquine in this process may be more complex than indicated by the data in this manuscript. Also, in this setting, if the authors hypothesis is correct that TLR8 is an important inflammatory axis in psoriasis, then chloroquine would be considered to have a therapeutic effect in psoriasis. However, I don't think this is the case and most reports suggest that chloroquine makes psoriasis worse. I'd like the authors to comment on this.

2. All the statistics shown in images are based on one-way ANOVA. Were the values in each group normally distributed? Also, as one-way ANOVA is an omnibus test statistic it cannot tell you which specific groups were statistically significantly different from each other, only that at least two groups were. Given the way and significance is displayed in the figures I'm not sure the one-way ANOVA is the correct statistical test to use in this setting. I'd like the authors to comment on this.

3. The size of the psoriasis group is very small (n=3). Although the authors describe this as significant (with one-way ANOVA test) this is not very robust, and the differences between the groups appear to be fairly minimal and of questionable biological significance. The authors can improve on this by

increasing their samples size.

4. The Transwell data to MIP-1beta and IL-16 are not very robust and do not appear to be significantly different, it is not apparent to me either that the "decrease" the authors state with the higher dose of each is actually lower.

5. I have some concern regarding the staining in Figure 5E. In Psor1 and Psor3 samples there is strong staining of Neutrophil Elastase throughout the upper layer of the epidermis – this is not how neutrophil infiltration would happen in psoriasis – it is not this diffuse, instead it is in localized small pockets or microabscesses – I'm concerned that this is staining artifacts - instead the Psor5 example is more what I'd expect with more focal collection of neutrophils. However, the orientation of that sample is suboptimal as the epidermis is at a 45-degree angle. The authors should consider fixing their images and replace their images with better examples (at least for Psor1 and Psor 3). Also, where is Psor2 and Psor4?

6. The inhibitory oligodeoxynucleotide experiments are of interest but it is not clear to me why the authors did not include more controls, i.e. knock-out lines of TLR7 and TLR8 – this would have made the data clearer. Also, do HEK293T cells express either TLR7 or TLR8 at baseline? How much did the transfection increase the levels of TLR7 and TLR8? Was this confirmed by WB? Such information should be provided in this manuscript. Why wasn't this experiment done with RNA+LL37? It's not clear to me why the authors instead used RNA complexed with DOTAP.

7. I have some concern regarding the RNA staining (Syto RNaselect) - it matches almost perfectly the DNA (Hoechst) staining and the Syto RNA-select staining is known to bind to DNA and fluoresce (although more weakly than when it is bound to RNA). This is exemplified in the data shown in S5 where the fluorescent is not gone in the presence of RNase treatment, it is merely somewhat suppressed. The authors should have included in that experiment a control with no nucleic acids.

8. The authors demonstrate that RNA is complexed within NETs along with LL37. Within the NETs I'd expect that the RNA+LL37 would be relatively walled-off and inactive and the NETs would have to be degraded for the RNA+LL37 to be released. Do the NETs have a direct activation effect on Neutrophils that is dependent upon RNA and/or LL37?

Reviewer #3, expert on innate immunity (Remarks to the Author):

The manuscript by Herster et al reports experiments revealing that human PMNs respond to complexes of RNA with cathelicidin peptide LL37 resulting in: 1) release of cytokines and chemokines, and 2) NETosis, with release of LL37, DNA, and RNA. The data support a model whereby in psoriasis PMNs stimulated by RNA-LL37 complexes contribute to inflammation, chemoattraction of immune cells and generation of a self-propagating proinflammatory cycle. This model suggests new potential targets for therapy. The study is interesting, the data are of high quality and the paper is well-written. I think it will certainly influence thinking in the field.

There are several items that the authors should address:

Direct evidence of TLR8 mediating the response to RNA-LL37 is somewhat limited. Thus, the abstract (line 49) might be an over-statement ("respond via TLR8"). Similarly, the title implies a TLR8 mediated response. On line 133, the authors state "we conclude TLR8 to be the receptor mediating the observed effects in PMNs", but direct evidence seems lacking. Related to this point, in experiments of Fig 6 (line 249), why was DOTAP used rather than LL37? If DOTAP is necessary for uptake in the HEK cells, might addition of LL37 enhance the response?

Line 258. It seems odd that IRS869 is not mentioned.

Figure 4A-C and line 191, "led to a donor dependent increase." Are the data for the low concentration simulation significant? The legend says $p > 0.05$. Would "paired-sample statistics" be appropriate? Minor, I did not see the concentration of SDF1a in legend or text, and the actual concentration of IL16 and MIP1b would improve the clarity of Fig4 legend.

Fig1F. Is a bar justified when $n=2$?

Fig 1G. It would be informative to know what proportion/% the selected cells approximate in the sample.

Fig 3A legend. Please define the arrow.

Fig 6A-C legend. Please define the arrows.

Several statements should be revised for clarity: Lines 41, 88, 242, 272

Universität Tübingen[®] Interfakultäres Institut für Zellbiologie
Abt. Immunologie [®] Auf der Morgenstelle 15 [®] 72076 Tübingen

We thank the Editor for the chance to appeal and append below a succinct response to the reviewer comments. We would also like to point out the addition of new data in Figs. 3D, I-M and Fig. S4A-E, all showing new *in vivo* data and/or additional controls. Regarding novelty we would like to highlight this is the first ever description of NET-associated RNA (naRNA).

Point-by-point reply

Below we append a short point-by-point reply to the previous reviewer comments.

Reviewer 1, expert on neutrophils

Comment 1: The authors present Figure 1) to argue that RNA-LL37 stimulates IL-8 cytokine release and the shedding of 62L marker from healthy neutrophils. However, the response is small (less than an RNA oligo R848 or other positive controls can achieve) and the CD62L shedding is a relatively subtle effect. So, the initial data in this report are nearly at background levels except for the uptake of label by PMN, which was anticipated from the known binding of LL37 to RNA.

Author reply #1: We appreciate the reviewer's main concern is the size of the effects. Admittedly, in Fig. 1A, the mean IL-8 output with RNA+LL37 is only >2-fold over background. But comparing Fig. 2E for another cytokine such as MIP-1 β , the ratio is >3-fold above RNA and >8-fold above background. In both cases, statistics unequivocally confirm the effect which should be taken into consideration. Furthermore RNA-LL37 induces clear CD62L shedding (Fig. 1C) that is not seen for R848 (Fig. 1A). Subsequently, we show that RNA-LL37 is a potent NET inducer using microscopy, compared to PMA. Thus, the activation of PMN by RNA-LL37 is documented by multiple methods, supported by statistics and depending on the readout comparable or exceeding well-established control stimuli.

Comment 2: In Figure 2), the secretion of TNF, IL-1 and IL-6, MIP1b and IL-16 are shown but the samples indicate that only certain few donors have a strong response, whereas most samples are not above background.

Author reply #2: As the reviewer is probably well aware, donor-to-donor variation is common when working with primary human neutrophils (PMNs), one of the most physiologically relevant and technically challenging systems. That not all donors respond equally is also seen for LPS and other stimuli and thus considered normal. Nevertheless statistics clearly show that the effects are real and not attributable to chance, despite the typically high donor-to-donor variation. Although this part of the analysis was more to

Department of Immunology

Prof. Dr. Hans-Georg Rammensee
Direktor

Office Management

Lynne Yakes
Tel.: +49 7071 29-87628
Fax: +49 7071 29-5653
E-mail: lynne.yakes@uni-tuebingen.de
<http://www.immunology-tuebingen.de/>

Gesprächspartner/Your contact person:

Prof. Dr. Alexander Weber
Professor of Innate Immunity

Tel.: +49 7071 29-87623
Mobil: +49-173 215 7220
Fax: +49 7071 29-4759

Tübingen, 19 July 2019

screen for which cytokines could be produced at all, we are happy to sample additional donors to improve the figure if necessary.

Comment 3: In Figure 3), samples of PMN from psoriasis patients are assayed but again the results are not very striking.

Author reply #3: We concede that the number of donors is limited. Despite our most intense efforts, we were unable to recruit additional patients with PASI >10 and without systemic treatment via the Tübingen, Mainz and Heidelberg psoriasis clinics. This is owed to the fact that many patients with this severity of disease are now treated systemically. We apologize that due to changing treatment standards further donors could not be included. Nevertheless, in life size research involving diseased patients, group sizes of 3 are not uncommon and 3 biological replicates allow for statistical analyses to be conducted. This analysis provides a 95% certainty for the conclusions drawn as is common practice. We therefore would like the editor and reviewers to take the number of patients analyzed here to gain a first impression on the clinical significance for psoriasis patients which clearly warrants further analysis. On the other hand, the main points of our study do not depend on the low number of psoriasis patients analyzed. To acknowledge the low number of subjects, we have toned down our statements about the effects on cytokine release in psoriasis patients in results, discussion and abstract order to address this point. Furthermore, we plan to measure additional donors and this data could be added during the revision process.

Comment 4: The Figure in which RNA is measured by SYTO RNaselect, concern that some of the signal is contributed by DNA cannot be avoided because the manufacturer indicates a weak binding to DNA.

Author reply #4: We appreciate and thank the reviewer for this concern which had initially also occurred to us. A very low binding to DNA has indeed been reported but Fig. S3A, B clearly show that RNase disparately affects the RNA signal (significantly reduced, S3A) and DNA signal (not affected, S3B). Other studies cited by us also confirm the specificity of the dye. To address this we have performed staining with anti-pseudo-uridine Abs (Ψ U is a nucleotide modification only found in RNA) that only detect RNA with virtually identical results. The anti- Ψ U signal was also demonstrated to be RNase but not DNase sensitive (Fig. 3D). The additional use of labeled RNA in Fig. S3 further corroborates the specificity of RNA staining.

Comment 5: In sum, the study is somewhat original but the overall purpose, to provide a major pathogenic mechanism for psoriasis, is not fully achieved. Instead, the study is consistent with previous knowledge and demonstrates a weak, but potentially meaningful, involvement of RNA-peptide complexes as further stimuli to amplify the innate immune response in psoriasis.

Author reply #5: Previously, LL37-RNA recognition by pDC (Gilliet et al) has received major attention but there have always been 4 major caveats: 1) What is the source of the nucleic acids? 2) How do they actually get released? 3) How come pDC supposedly play a major role if they are only a minor cell population in psoriasis skin and blood? 4) pDC do not produce inflammatory cytokines in high amounts, yet psoriasis is clearly driven by these cytokines. For the first time, our paper connects these mechanistically highly relevant findings and provides an attractive unified scenario that includes a self-exacerbating progression towards skin inflammation which we now demonstrate *in vivo*. Furthermore, we consider the first description of NET-associated RNA (naRNA) as a significance advance. In light of multiple diseases with anti-RNA autoantibodies this finding will open up a new angle in both NET and autoimmunity research.

Reviewer 2, expert on psoriasis

Comment 6: The chloroquine inhibition of response to RNA-LL37 – led the authors to conclude that TLR8 was the receptor mediating the observed effects in PMNs. The authors also show co-localization of RNA40 with TLR8 in PMNs (Figure 1H). The role of TLR8 in this process is of interest but I do think the authors need to demonstrate this more conclusively by using siRNA approaches etc. I realize that PMNs can be hard to work with but I think this is essential, particularly as some groups have shown that chloroquine may augment activation of synthetic compounds such as R848 (Kuznik A, JI 2011).

Author reply #6: We know the Kuznic 2011 study very well but the effects of chloroquine were assessed only for small compounds (e.g. R848, Fig. 2C of this paper) or DNA oligos but not RNA (see Fig. 2C). In order to not rely only on this dataset for naming a receptor and to follow the reviewer's suggestions, we have now conducted work in BM-PMN from mice deficient in TLR13, the functional equivalent of human TLR8 in mice (Li and Chen 2012; Eigenbrod, Pelka et al. 2015; Hafner, Kolbe et al. 2019), showing impairment of both cytokine and NET response to RNA-LL37 but not PMA (new Figs. 4A, C). These observations are in line with the earlier data on chloroquine (Fig. 1I), Unc93B1 KOs (Fig. 4B) and inhibitory oligos (Fig. 4F, G). We have also used TLR8-CRISPRed macrophage cells to show in the human system that TLR8 detects RNA-LL37 complexes (new Fig. 4D). That PMN do not express TLR7 was shown before (Berger, Hsieh et al. 2012) and we confirmed in our hands this in that human primary PMN or mouse BM-PMN do not respond to the TLR7-exclusive stimulus imiquimod. We have not added this negative data to the manuscript but would be prepared to do so if the reviewer or editor considers this helpful.

Comment 7: Therefore, the role of chloroquine in this process may be more complex than indicated by the data in this manuscript. Also, in this setting, if the authors hypothesis is correct that TLR8 is an important inflammatory axis in psoriasis, then chloroquine would be considered to have a therapeutic effect in psoriasis. However, I don't think this is the case and most reports suggest that chloroquine makes psoriasis worse. I'd like the authors to comment on this.

Author reply #7: We appreciate the reviewer's comment and agree that one would theoretically expect a beneficial effect of chloroquine in psoriasis. Generally, it is difficult to extrapolate from our observations in vitro on PMN to psoriasis, to clinical studies as treatment would naturally occur during late phase disease where the disease is driven by T cells rather than PMNs. Several older studies (mainly case reports) admittedly seem to link chloroquine with psoriasis worsening but even early studies have seen the opposite effect (O'Quinn, Kennedy et al. 1964). In a 1970s study involving 100 patients, no adverse effects of chloroquine were observed and several patients experienced a remission. Due to these conflicting reports several later authors have questioned the early conclusions that chloroquine exacerbates psoriasis (Sorbara, Cozzani et al. 2006). Even authors who note the reported link between chloroquine and psoriasis concede that "well-conducted systematic studies on drug-related psoriasis are mostly lacking" (Balak and Hajdarbegovic 2017). As a matter of fact, to the best of our knowledge there have been no registered well-controlled large-scale trials of chloroquine and most reports are case reports of <20 patients. Thus the jury is still out on the effect of chloroquine treatment in psoriasis. Generally, the clinical picture of chloroquine treatment in psoriasis should not be taken to support or discount our present study where it was exclusively used in vitro as an inhibitor of endosomal TLR signaling (used in many studies). We would be

happy to add the abovementioned references to illustrate this controversy to the reader but feel this would distract from the main points of the paper.

Comment 8: 2. All the statistics shown in images are based on one-way ANOVA. Were the values in each group normally distributed? Also, as one-way ANOVA is an omnibus test statistic it cannot tell you which specific groups were statistically significantly different from each other, only that at least two groups were. Given the way and significance is displayed in the figures I'm not sure the one-way ANOVA is the correct statistical test to use in this setting. I'd like the authors to comment on this.

Author reply #8: As stated in methods we always checked the distribution of values in each group. Only when there was normal distribution, an ANOVA was used, otherwise parametric tests (e.g. Friedman). Of course, we used multiple testing and report here adjusted p-values for individual comparisons, not the overall ANOVA statistics which indeed are not helpful. We have re-checked the statistics description in the methods section and hope to have clarified this misunderstanding.

Comment 9: 3. The size of the psoriasis group is very small (n=3). Although the authors describe this as significant (with one-way ANOVA test) this is not very robust, and the differences between the groups appear to be fairly minimal and of questionable biological significance. The authors can improve on this by increasing their samples size.

Author reply #9: See author reply #3.

Comment 10: 4. The Transwell data to MIP-1beta and IL-16 are not very robust and do not appear to be significantly different, it is not apparent to me either that the "decrease" the authors state with the higher dose of each is actually lower.

Author reply #10: We concede that due to the high donor-to-donor variation (See above) the differences are not statistically significant (as we point out to the reader to avoid overinterpretation). Since it is not essential for the main message of our paper, we are happy to remove this data, conceding that the effects are modest.

Comment 11: 5. I have some concern regarding the staining in Figure 5E. in Psor1 and Psor3 samples there is strong staining of Neutrophil Elastase throughout the upper layer of the epidermis – this is not how neutrophil infiltration would happen in psoriasis – it is not this diffuse, instead it is in localized small pockets or microabscesses – I'm concerned that this is staining artifacts - instead the Psor5 example is more what I'd expect with more focal collection of neutrophils. However, the orientation of that sample is suboptimal as the epidermis is at a 45-degree angle. The authors should consider fixing their images and replace their images with better examples (at least for Psor1 and Psor 3). Also, where is Psor2 and Psor4?

Author reply #11: Figure 5: The staining the upper layer of the stratum corneum, that occurs in multiple channels, is clearly autofluorescence and we regret not having labeled it as such. The remaining signal for NE corresponds to what is consistent with the literature and the reviewer's expectation. The 45° angle was due to the orientation of the sample on the slide but can easily be adjusted. A number of additional samples have also been measured since (total of 12 patients and 3 healthy controls). Generally, it would seem commonplace that only a selection of donors is shown due to space restrictions. We have ameliorated the numbering of patient samples to avoid confusion.

Comment 12: 6. The inhibitory oligodeoxynucleotide experiments are of interest but it is not clear to me why the authors did not include more controls, i.e. knock-out lines of TLR7 and TLR8 – this would have made the data clearer. Also, do HEK293T cells express either TLR7 or TLR8 at baseline? How much did the transfection increase the levels of TLR7 and TLR8? Was this confirmed by WB? Such information should be provided in this manuscript. Why wasn't this experiment done with RNA+LL37? It's not clear to me why the authors instead used RNA complexed with DOTAP.

Author reply #12: That HEK293T cells do not express TLR7 or TLR8 and only become responsive to ligands upon transfection has been shown numerous times, e.g. (Colak, Leslie et al. 2014), where the same constructs were used and immunoblots with and without transfection are shown. We could easily demonstrate expression here. The experiment was done with DOTAP for convenience sake but could be easily re-done with LL37. We did not consider this necessary, as the iODN also worked even more effectively on the more physiologically much more relevant PMNs stimulated with RNA-LL37 complexes (Fig. 4F-I), and at much lower iODN concentrations (only 1-50 nM). To not distract from the PMN data, most HEK293T experiments have now been relegated to the supplement in response to the reviewer's comment. Additionally, we have now included data from cell lines lacking TLR8, see reply #6.

Comment 13: 7. I have some concern regarding the RNA staining (Syto RNaselect) - it matches almost perfectly the DNA (Hoechst) staining and the Syto RNA-select staining is known to bind to DNA and fluoresce (although more weakly than when it is bound to RNA). This is exemplified in the data shown in S5 where the fluorescent is not gone in the presence of RNase treatment, it is merely somewhat suppressed. The authors should have included in that experiment a control with no nucleic acids.

Author reply #13: Please see reply #4 regarding additional staining and controls. It is actually conceivable that the extruded RNA stems from nucleoli which contain small nucleolar RNA (snRNA) – something to be analyzed based on our report.

Comment 14: 8. The authors demonstrate that RNA is complexed within NETs along with LL37. Within the NETs I'd expect that the RNA+LL37 would be relatively walled-off and inactive and the NETs would have to be degraded for the RNA+LL37 to be released. Do the NETs have a direct activation effect on Neutrophils that is dependent upon RNA and/or LL37?

Author reply #14: We can now demonstrate a direct activation effect of NETs on naïve PMNs (new Fig. 3L and M). As shown in the IF images of Fig. 3C, RNA and LL37 are distributed along the NETs and are accessible to antibody staining (anti-ΨU, Fig. 3D and 3K, Fig. S4D) so that degradation for release is probably not necessary.

Reviewer 3, expert on innate immunity

Comment 15: Direct evidence of TLR8 mediating the response to RNA-LL37 is somewhat limited. Thus, the abstract (line 49) might be an over-statement ("respond via TLR8"). Similarly, the title implies a TLR8 mediated response. On line 133, the authors state "we conclude TLR8 to be the receptor mediating the observed effects in PMNs", but direct evidence seems lacking. Related to this point, in experiments of Fig 6 (line 249), why was DOTAP used rather than LL37? If DOTAP is necessary for uptake in the HEK cells, might addition of LL37 enhance the response?

Author reply #15: We appreciate the reviewer's concern to avoid overinterpretation and have therefore addressed the question of TLR8, see reply #6, and DOTAP, see reply #12. We feel that our data from Unc93B1 and TLR13 KO BM-PMNs and from TLR8 CRISRPed macrophage cells (Vierbuchen, Bang et al. 2017) are sufficient to conclude that TLR8 is the receptor in humans, TLR13 in mice, consistent with the role of these receptors in bacterial RNA recognition (Eigenbrod, Franchi et al. 2012; Hafner, Kolbe et al. 2019).

Comment 16: Line 258. It seems odd that IRS869 is not mentioned.

Author reply #16: We acknowledge this small inconsistency. Due to sometimes limited numbers of cells and the availability of reagents, the analysis was limited to only IRS661 and IRS954. If considered mandatory, further analyses with the abovementioned IRS869 could be conducted.

Comment 17: Figure 4A-C and line 191, "led to a donor dependent increase." Are the data for the low concentration simulation significant? The legend says $p > 0.05$. Would "paired-sample statistics" be appropriate?

Author reply #17: We thank the reviewer for this suggestion and refer to reply #10. We had used a paired test as suggested by the reviewer and have explicitly added this to the figure legend.

Comment 18: Minor, I did not see the concentration of SDF1a in legend or text, and the actual concentration of IL16 and MIP1b would improve the clarity of Fig4 legend.

Author reply #18: We thank the reviewer for this suggestion and have added the concentrations to the Figure 2 legend. The concentration of SDF was 100 ng/ml and this was added to the supplemental figure legend (Fig. S2C-E).

Comment 19: Fig1F. Is a bar justified when $n=2$?

Author reply #19: The reviewer is correct in that the calculation of a standard deviation is not relevant for $n=2$. We have removed the error bar accordingly.

Comment 20: Fig 1G. It would be informative to know what proportion/% the selected cells approximate in the sample.

Author reply #20: We thank the reviewer. At least 10.000 cells were acquired for each sample with 40x magnification using an ImageStream X MKII with the INSPIRE instrument controller software (Merck-Millipore/Amnis). Only cells which were in focus were measured. We additionally stained CD15 as a PMN marker. All measured CD15+ positive cells per sample were counted as 100 %. The information has also been added to the methods section.

Comment 21: Fig 3A legend. Please define the arrow.

Author reply #21: We refer to the arrow in the Results section. That this is the intention has been added to the Fig. 2 legend.

Comment 22: Fig 6A-C legend. Please define the arrows.

Author reply #22: The arrows indicate the level of receptor stimulation in the absence of any IRS. We apologize for the omission. This has been made clear in the figure legends (now Fig. S4A-C).

Comment 23: Several statements should be revised for clarity: Lines 41, 88, 242, 272

Author reply #23: We thank the reviewer for these suggestions which have been ameliorated.

Cited references

- Balak, D. M. and E. Hajdarbegovic (2017). "Drug-induced psoriasis: clinical perspectives." *Psoriasis (Auckl)* 7: 87-94.
- Berger, M., C. Y. Hsieh, et al. (2012). "Neutrophils express distinct RNA receptors in a non-canonical way." *The Journal of biological chemistry* 287(23): 19409-19417.
- Colak, E., A. Leslie, et al. (2014). "RNA and imidazoquinolines are sensed by distinct TLR7/8 ectodomain sites resulting in functionally disparate signaling events." *J Immunol* 192(12): 5963-5973.
- Eigenbrod, T., L. Franchi, et al. (2012). "Bacterial RNA mediates activation of caspase-1 and IL-1beta release independently of TLRs 3, 7, 9 and TRIF but is dependent on UNC93B." *J Immunol* 189(1): 328-336.
- Eigenbrod, T., K. Pelka, et al. (2015). "TLR8 Senses Bacterial RNA in Human Monocytes and Plays a Nonredundant Role for Recognition of *Streptococcus pyogenes*." *J Immunol* 195(3): 1092-1099.
- Hafner, A., U. Kolbe, et al. (2019). "Crucial Role of Nucleic Acid Sensing via Endosomal Toll-Like Receptors for the Defense of *Streptococcus pyogenes* in vitro and in vivo." *Front Immunol* 10: 198.
- Li, X. D. and Z. J. Chen (2012). "Sequence specific detection of bacterial 23S ribosomal RNA by TLR13." *Elife* 1: e00102.
- O'Quinn, S. E., C. B. Kennedy, et al. (1964). "Psoriasis, Ultraviolet Light, and Chloroquine." *Arch Dermatol* 90: 211-216.
- Sorbara, S., E. Cozzani, et al. (2006). "Hydroxychloroquine in psoriasis: is it really harmful?" *Acta Derm Venereol* 86(5): 450-451.
- Vierbuchen, T., C. Bang, et al. (2017). "The Human-Associated Archaeon *Methanosphaera stadtmanae* Is Recognized through Its RNA and Induces TLR8-Dependent NLRP3 Inflammasome Activation." *Front Immunol* 8: 1535.

Reviewers' comments:

Reviewer #1 (Remarks to the Author):

The authors have provided reasonable rationale in response to the reviewers comments. Where possible, additional experiments were conducted. Control experiments, in general, confirm the initial observations. The revision appears to adequately address previous comments.

Reviewer #2 (Remarks to the Author):

The authors have addressed my comments and critiques. I have nothing further to add.

Reviewer #3 (Remarks to the Author):

The revised manuscript by Herster et al reports experiments revealing that human PMNs respond to complexes of RNA-LL37 that result in: 1) release of cytokines and chemokines, and 2) NETosis, with release of LL37, DNA, and RNA. The manuscript newly provides evidence that NETs contain RNA. The data support a model whereby in psoriasis PMNs stimulated by RNA-LL37 complexes contribute to inflammation, chemoattraction of immune cells and generation of a self-propagating proinflammatory cycle. The model suggests new potential targets for therapy. The study is interesting, the data are of high quality (with a few points needing clarification below) and the paper is well-written. I think the study will influence thinking in the field.

There remain several items that the authors should address:

Figure 2. The data for IL-16 including 2D, S2B, 2F, 2G, 2H are not impressive, especially compared to the positive control (SDF-1). The data for MIP-1b are a bit better, at least the expression data (2E and S2B). How important to their story do the authors view these data? It seems that 2A + 2B + 2C do an adequate job of supporting line #195: "We conclude that the combination of RNA with LL37 triggers the release of an extended array of pro-inflammatory cytokines and chemoattractants not triggered by RNA alone." Would it be better to down-play and move the less convincing data to the supplement?

Figure 2 J and K. I am not an expert in statistics, but the authors seem to be drawing conclusions on more than one comparison:

"both IL-8 and MIP-1 β release was significantly increased two-fold in response to RNA-LL37 (Fig. 2J, K)"

and

"psoriasis PMNs treated with 'RNA alone' (Fig. 2J, arrow) produced significantly more IL-8 than both 'unstimulated' psoriasis PMNs and healthy donor PMNs stimulated with 'RNA alone'"

Would a two-ANOVA be more appropriate?

Lines 223-226. I do not see any of these figures or movies:

"To further exclude artefacts that might arise from staining fixed samples, RNA-LL37 mediated NETosis was also analyzed and confirmed using live-cell time-lapse analysis of PMNs (Fig. 3E and Movies S1-S3, quantified in Fig. 3F). Interestingly, in PMNs captured on the verge of NETosis, RNA staining in a granula-like fashion could be clearly observed (Fig. S3F)."

Line 232. I was not provided with Movie S4.

Fig 4E and Suppl Figs 5 A, B, C, D, E. There is a formatting issue. The x-axis labels are seriously askew making these panels very difficult to interpret. Also, the text describing these experiments (lines 279-299) is hard to follow and may benefit from revision.

Fig 4F. I worry that 4 of 6 data points for RNA+LL37 (3rd bar) do not seem to show much difference than data in the presence of IRS661 (4th bar) or in the presence of IRS954 (5th bar). The text seems oddly phrased in describing this panel and the p value in the figure is marginally significant. The data for MIP1b (Fig 4G) support better the claims. Granted the samples are clinical specimens with likely inherent variability, so consider just revising the text description of results in 4F to be more straightforward (and maybe consider presenting MIP1b before the marginally significant IL8 results).

Figure 1, figure and legend. Shouldn't the Y-axis for "E", right panel (flow plot) be "AF647". The legend information following "(E-G)" should be revised. I think it should be "(E-H)", and the scale bars are for G & H (not F). Consider, "in G five selected cells..." rather than, "in G selected cells..."

Line 218. I think the call out should be 3E, not 3D.

Line 219. "...was completely RNase sensitive." Should a call out to 3D be made here?

Line 279. Consider: "Having identified TLR8 as the likely responsible receptor in humans..."

Detailed point-by-point reply NCOMMS-18-24590B

We like to thank the Editor for supporting our request to appeal and the opportunity to respond to the latest reviewer comments in this point-by-point reply. We hope this addresses all remaining concerns adequately. All changes have been highlighted.

Below we append a short point-by-point reply to the previous reviewer comments.

Reviewer 1

Comment 1: Reviewer #1 (Remarks to the Author):

The authors have provided reasonable rationale in response to the reviewers comments. Where possible, additional experiments were conducted. Control experiments, in general, confirm the initial observations. The revision appears to adequately address previous comments.

Author reply #1: We thank reviewer 1 for this positive feedback.

Comment 2: Reviewer #2 (Remarks to the Author): The authors have addressed my comments and critiques. I have nothing further to add.

Author reply #2: We appreciate reviewer 2's favorable assessment.

Comment 3: Reviewer #3 (Remarks to the Author):

The revised manuscript by Herster et al reports experiments revealing that human PMNs respond to complexes of RNA-LL37 that result in: 1) release of cytokines and chemokines, and 2) NETosis, with release of LL37, DNA, and RNA. The manuscript newly provides evidence that NETs contain RNA. The data support a model whereby in psoriasis PMNs stimulated by RNA-LL37 complexes contribute to inflammation, chemoattraction of immune cells and generation of a self-propagating proinflammatory cycle. The model suggests new potential targets for therapy. The study is interesting, the data are of high quality (with a few points needing clarification below) and the paper is well-written. I think the study will influence thinking in the field. There remain several items that the authors should address:

Author reply #3: We thank reviewer 3 for her/his appreciation of our work and its potential and below address the items flagged up by the reviewer.

Comment 4: Figure 2. The data for IL-16 including 2D, S2B, 2F, 2G, 2H are not impressive, especially compared to the positive control (SDF-1). The data for MIP-1b are a bit better, at least the expression data (2E and S2B). How important to their story do the authors view these data? It seems that 2A + 2B + 2C do an adequate job of supporting line #195: "We conclude that the combination of RNA with LL37 triggers the release of an extended array of pro-inflammatory cytokines and chemoattractants not triggered by RNA alone." Would it be better to down-play and move the less convincing data to the supplement?

Author reply #4: We appreciate the reviewer's comments and have to concur that e.g. the IL-16 data are not as strong and possibly distract from the main thrust of the paper. We therefore toned the description down and moved the IL-16 data to the supplement (now Fig. S3C-F; note the resulting changes in panel numbering of Fig. 2 and S2) according to the reviewer's helpful suggestion.

Comment 5: Figure 2 J and K. I am not an expert in statistics, but the authors seem to be drawing conclusions on more than one comparison:

"both IL-8 and MIP-1 β release was significantly increased two-fold in response to RNA-LL37 (Fig. 2J, K)"
and

"psoriasis PMNs treated with 'RNA alone' (Fig. 2J, arrow) produced significantly more IL-8 than both 'unstimulated' psoriasis PMNs and healthy donor PMNs stimulated with 'RNA alone')"

Would an two-ANOVA be more appropriate?

Author reply #5: We apologize if the phrasing of the description was misleading. We agree with the reviewer that simultaneously comparing both "disease status" (patient vs healthy) and "response to treatment" (e.g. unstim vs RNA vs RNA+LL37 vs LL37) would require a two-way ANOVA setup. Using a two-way ANOVA the difference in response is significantly higher in patients vs controls only for RNA+LL37 but not any other stimuli (all of which are almost not eliciting a response above background). However, "disease" did not emerge as statistically different due to the fact that 3 out of 4 stimuli do not elicit a strong response. Thus, globally, patients and healthy donors, even though differing in their response to RNA+LL37 and of course clinically, are indistinguishable. This validates that the observation for RNA+LL37 is truly specific as observed for similar cytokine release in response to LPS (Fig. S2J, L). However, we felt that reporting "disease" as a variable that is *not* different might provide a confusing or contradictory message to the reader. We therefore decided not to include "disease" as a test variable, hence the one-way ANOVA. We apologize this was not entirely clear and have sought to clarify the wording of the revised results and legend to avoid misunderstandings and highlight only the "response to treatment" was tested. Furthermore, since in the IL-8 data RNA-LL37 did not reach significance (but showed the same trend), this dataset was now correctly described but relegated to supplement.

Comment 6: Lines 223-226. I do not see any of these figures or movies:

"To further exclude artefacts that might arise from staining fixed samples, RNA-LL37 mediated NETosis was also analyzed and confirmed using live-cell time-lapse analysis of PMNs (Fig. 3E and Movies S1-S3, quantified in Fig. 3F). Interestingly, in PMNs captured on the verge of NETosis, RNA staining in a granula-like fashion could be clearly observed (Fig. S3F)."

Author reply #6: We apologize if the impression arose that these files had not been included. We have re-checked and Fig. 3E, F and S3F were included in the submitted figure files (see low-resolution overview of the figures with panels highlighted on the next page; high quality figures uploaded) and represent quantifications of the live cell analysis, but we also apologize if the arrangement of panels has been confusing. We await comments from the editor or layout team whether this should be ameliorated. Since they had not been modified since before the appeal, Movies S1-S3 had mistakenly not been uploaded again in the appeal version. We have now uploaded the movies again and sincerely apologize that this meant the reviewer could not inspect these files. Movie S4 has also been uploaded.

Comment 7: Line 232. I was not provided with Movie S4.

Author reply #7: We sincerely apologize for failing to upload Movie S4 which has now been done. The movie represents a 3D reconstruction of the tissue analysis shown in Fig. 3H.

Comment 8: Fig 4E and Suppl Figs 5 A, B, C, D, E. There is a formatting issue. The x-axis labels are seriously askew making these panels very difficult to interpret. Also, the text describing these experiments (lines 279-299) is hard to follow and may benefit from revision.

Author reply #8: We apologize for this formatting issue which we have corrected. We have also sought to increase the clarity of the figure and have stream-lined the description as suggested by the reviewer.

Comment 9: Fig 4F. I worry that 4 of 6 data points for RNA+LL37 (3rd bar) do not seem to show much difference than data in the presence of IRS661 (4th bar) or in the presence of IRS954 (5th bar). The text seems oddly phrased in describing this panel and the p value in the figure is marginally significant. The data for MIP1b (Fig 4G) support better the claims. Granted the samples are clinical specimens with likely inherent variability, so consider just revising the text description of results in 4F to be more straightforward (and maybe consider presenting MIP1b before the marginally significant IL8 results).

Author reply #9: We thank the reviewer for this assessment and in the revised manuscript have therefore accordingly referred to MIP-1 β first and IL-8 in the supplemental data in order to lay the emphasis on the strongest data. Generally, the text description has also been revised to improve clarity.

Comment 10: Figure 1, figure and legend. Shouldn't the Y-axis for "E", right panel (flow plot) be "AF647". The legend information following "(E-G)" should be revised. I think it should be "(E-H)", and the scale bars are for G & H (not F). Consider, "in G five selected cells..." rather than, "in G selected cells..."

Author reply #10: We apologize the label for the Fig. 1E axis was missing and have re-inserted the AF647 label as the reviewer correctly pointed out was necessary. The figure legend has also been amended as suggested.

Comment 11: Line 218. I think the call out should be 3E, not 3D. Line 219. "...was completely RNase sensitive." Should a call out to 3D be made here?

Author reply #11: Indeed the callout should be made to Figure 3D at the end of the sentence. A callout to Fig. 3E was mistakenly added earlier in the sentence. We hope the description is now clearer and thank the reviewer for flagging this issue up.

Comment 12:

Line 279. Consider: "Having identified TLR8 as the likely responsible receptor in humans..."

Author reply #12: Although we feel that TLR8 data from human BlaER1 macrophages (Fig. 4D), the TLR13 data from mouse BM-PMNs (Fig. 4A) and the chloroquine (Fig. 1I) and iODN (Fig. 4E) effects in primary human PMNs, make the assumption that TLR8 is the receptor in primary human neutrophils highly plausible, we agree that at this stage we cannot demonstrate (e.g. genetically) that TLR8 is responsible in primary human PMNs. We therefore have taken the suggestion of reviewer 3 on board and have re-phrased the sentence accordingly.

Additional minor errors and wordings throughout the manuscript and figures were improved also. The abstract was also revised to comply with Nature Communications length restrictions (150 words only).

REVIEWERS' COMMENTS:

Reviewer #3 (Remarks to the Author):

The authors have addressed the questions and comments I raised to my satisfaction. I congratulate the authors on their exciting investigation that will influence thinking in the field. Well done!